https://doi.org/10.5194/egusphere-2023-103



# Thermal infrared dust optical depth and coarse-mode effective diameter retrieved from collocated MODIS and CALIOP observations

Jianyu Zheng[1,2], Zhibo Zhang[1,2,*], Hongbin Yu[3], Anne Garnier[4,5], Qianqian Song[1,2], Chenxi Wang[2,3], Claudia Di Biagio[6], Jasper F. Kok[7], Yevgeny Derimian[8], Claire Ryder[9]

[1]Department of Physics, University of Maryland Baltimore County, Baltimore, MD 21250, USA
[2]Goddard Earth Sciences Technology and Research II, University of Maryland Baltimore County, Baltimore, MD
21250, USA
[3]NASA Goddard Space Flight Center, Greenbelt, MD 20771, USA
[4]Science Systems and Applications, Inc., Hampton, VA, USA
[5]NASA Langley Research Center, Hampton, VA, USA
[6]Université Paris Cité and Univ Paris Est Creteil, CNRS, LISA, F-75013 Paris, France
[7]Department of Atmospheric and Oceanic Sciences, University of California, Los Angeles, CA 90095, USA
[8] French National Centre for Scientific Research | CNRS · Laboratoire d'optique atmosphérique (LOA)
[9] Department of Meteorology, University of Reading, Reading, UK.

*Correspondence to*: Zhibo Zhang (zzbatmos@umbc.edu)

**Abstract.**

In this study, we developed a novel algorithm based on the collocated Moderate Resolution Imaging Spectroradiometer (MODIS) thermal infrared (TIR) observations and dust vertical profiles from the Cloud-Aerosol Lidar with Orthogonal Polarization (CALIOP) to simultaneously retrieve dust aerosol optical depth at 10 μm ($DAOD_{10μm}$) and the coarse-mode dust effective diameter ($D_{eff}$) over global oceans. The accuracy of the $D_{eff}$ retrieval is assessed by comparing the PSD corresponding to retrieved $D_{eff}$ with the in-situ measured dust particle size 25    distributions (PSDs) from the AER-D, SAMUM-2 and SALTRACE field campaigns through case studies. The new $DAOD_{10μm}$ retrievals were evaluated first through comparisons with the collocated $DAOD_{10.6μm}$ retrieved from the combined Imaging Infrared Radiometer (IIR) and CALIOP observations from our previous study (Zheng et al. 2022). The pixel-to-pixel comparison of the two retrievals indicates a good agreement (R~0.7) and a significant reduction of (~50%) retrieval uncertainties largely thanks to the better constraint on dust size. In a climatological comparison, the 30    seasonal and regional (5°×2°) mean $DAOD_{10um}$ retrievals based on our combined MODIS and CALIOP method are in good agreement with the two independent Infrared Atmospheric Sounding Interferometer (IASI) products over three dust transport regions (i.e., North Atlantic (NA; R = 0.9), Indian Ocean (IO; R = 0.8) and North Pacific (NP; R = 0.7)).

    Using the new retrievals from 2013 to 2017, we performed a climatological analysis of coarse mode dust $D_{eff}$ over 35    global oceans. We found that dust $D_{eff}$ over IO and NP are up to 20% smaller than that over NA. Over NA in summer, we found a ~50% reduction of the number of retrievals with $D_{eff} > 5$ μm from 15°W to 35°W and a stable trend of $D_{eff}$ average at 4.4 μm from 35°W throughout the Caribbean Sea (90°W). Over NP in spring, only ~5% of retrieved pixels with $D_{eff} > 5$ μm are found from 150°E to 180°, while the mean $D_{eff}$ remains stable at 4.0 μm throughout eastern NP. To our best knowledge, this study is the first to retrieve both DAOD and coarse-mode dust particle size over global





oceans for multiple years. This retrieval dataset provides insightful information for evaluating dust long-wave radiative effects and coarse mode dust particle size in models.

## 1 Introduction

Mineral dust (referred to as "dust") lifted by strong surface winds in arid and semi-arid regions (Ginoux et al., 2012) is the most abundant type of atmospheric aerosol in terms of dry mass (Kinne et al., 2006; Goudie, 1983). Once aloft,

dust particles can be carried by winds for long-range transport on intercontinental to hemispherical scales, exerting far-reaching impacts on the climate system (Shao et al., 2011; Choobari et al., 2014; Yu et al., 2013; Tegen and Fung, 1994; Uno et al., 2009). For example, dust significantly influences the earth system's radiative budget by interacting with both shortwave (SW) solar and longwave (LW) terrestrial radiations, known as the direct radiative effects (DRE). Previous studies have found that on a global mean basis, the dust DRE at the top of the atmosphere (TOA) is generally

negative in SW (i.e., a cooling effect) but positive (i.e., a warming effect) in LW, although dust SW DRE can be positive over bright surfaces (Kok et al. 2017, Li et al. 2021, Di Biagio et al., 2021, Song et al. 2018, 2022). Despite this qualitative understanding, the quantification of dust net DRE (i.e., SW DRE + LW DRE) remains highly uncertain, in part due to the great spatiotemporal heterogeneity of dust properties such as dust loading, optically represented by dust aerosol optical depth (DAOD) (Huneeus et al., 2011), particle size distribution (PSD) (Kok et al. 2017, Adebiyi

and Kok, 2020a), particle shape, and refractive indices (RI) (Li et al., 2021). The dust-radiation interactions perturb the surface energy balance and atmospheric heating rate and hence the thermodynamic structure of the atmosphere (Helmert et al., 2007), which in turn affects boundary layer dynamics and cloud formation processes (Amiri-Farahani et al., 2017; Grogan et al., 2016; Zhang et al., 2007).

Satellite remote sensing is uniquely capable of measuring the spatiotemporal variation of dust properties on regional to global scales and over years and decades. Many methods have been developed to retrieve the column-integrated AOD in the visible spectrum ($AOD_{VIS}$) (e.g., 550 nm) from passive satellite observations in the visible (VIS) and near-infrared (NIR) spectrum, such as the Moderate Resolution Imaging Spectroradiometer (MODIS) (Levy et al., 2013, Hsu et al., 2013) and the Multi-angle Imaging Spectroradiometer (MISR) (Kahn et al., 2010). It should be noted that

these retrievals obtain the total AOD contributed by not only dust but also other types of aerosols. As a result, the fraction of dust AOD (DAOD) in VIS ($DAOD_{VIS}$) needs to be further separated from the total $AOD_{VIS}$ for dust-focused studies. Some methods rely on model simulations of $DAOD_{VIS}/AOD_{VIS}$ (Gkikas et al., 2021) or non-dust $AOD_{VIS}$ (Ridley et al., 2016). Others are based on the contrasting properties of dust in comparison with other aerosols, such as its larger size manifested as a smaller Angstrom Exponent and a smaller fine-mode fraction (Kaufman et al., 2005, Yu

et al., 2009, 2021) and significant spectral gradient in the absorption from deep blue to the VIS (Ginoux et al., 2010, Pu and Ginoux, 2018). In addition, the active spaceborne Lidars with VIS-NIR channels, such as the Cloud-Aerosol Lidar with Orthogonal Polarization (CALIOP) onboard the Cloud-Aerosol Lidar and Infrared Pathfinder Satellite Observations (CALIPSO) mission and the Cloud-Aerosol Transport System (CATS), can be used to estimate the vertical distribution of $DAOD_{VIS}$ based on the observed particulate depolarization ratios (Yu et al., 2015b; Proestakis

et al., 2018). By utilizing these retrieval methods, several studies further developed decade-long satellite data records



of DAOD$_{VIS}$ (Gkikas et al., 2022; Song et al., 2021), which are frequently used for dust studies such as estimations of dust DRE, interannual variability and trends of dust, and global dust cycles (Song et al., 2022; Logothetis et al., 2021; Kok et al., 2021b).

The VIS-NIR dust observations are useful, but they do not provide direct measurements of DAOD at LW and have weak sensitivity to coarse particles (Ryder et al., 2019). Extending observed DAOD from the VIS-NIR to TIR spectra depends strongly on dust PSD and RI assumptions (Song et al., 2018). Therefore, TIR observations are an indispensable complement with several unique advantages. Dust dominated by coarse mode particles is arguably the only predominant particle that can cause strong radiative signatures in the TIR spectrum (Desouza-Machado et al.,
2006). Therefore, using TIR observation has an inherent advantage of directly retrieving DAOD, in contrast to the VIS-NIR observations in which empirical methods need to be developed to separate dust from other types of aerosols. Zheng et al. (2022) showed that direct TIR observations could significantly reduce uncertainties in DAOD$_{TIR}$ and LW DRE associated with dust PSD and RI assumptions. Moreover, previous studies revealed that super-coarse dust particles ($D_p > 20$ μm) are ubiquitously detected from numerous in-situ measurements in both source regions and
transport regions (Weinzierl et al., 2017; Denjean et al., 2016; Ryder et al., 2013b; Ryder et al., 2018), which however is excluded or underestimated in most dust transport models (Checa-Garcia et al., 2021; Wu et al., 2020; Zhao et al., 2022). How many and how frequently can super-coarse dust particles be carried in long-range transport? The lack of observational data with finer spatiotemporal coverage prevents us from further revealing their transport patterns. TIR satellite observations with great sensitivity to super coarse particles can potentially fill this knowledge gap.


Notwithstanding the advantages, retrieving dust properties in TIR, particularly the dust particle size, usually represented by effective radius or diameter, is challenging. In the past, TIR dust retrieval algorithms were primarily based on observations from space-borne hyperspectral atmospheric sounders, such as the Advanced Infrared Radiation Sounder (AIRS) and the Infrared Atmospheric Sounding Interferometer (IASI). The important advantages of
hyperspectral observations for dust retrieval are that they can provide multiple atmospheric window channels that are most sensitive to dust aerosols and less sensitive to atmospheric gas absorption (Peyridieu et al., 2010; Capelle et al., 2018; Capelle et al., 2014; Peyridieu et al., 2013). On the other hand, these algorithms have two major limitations. First, the altitude of a dust layer and, therefore, its temperature profile affect the outgoing TIR radiance at TOA with a similar magnitude as DAOD (Pierangelo et al., 2004). As a result, dust altitude must be part of the state vector to be
retrieved together with DAOD in a stand-alone hyperspectral TIR dust retrieval algorithm, which makes retrieving dust particle size from limited information contents highly challenging (Pierangelo et al., 2005). Second, the relatively large footprint of hyperspectral sounders (~15 km) makes cloud masking and clearing a daunting task. As a result, the retrieval results are prone to cloud contamination (Song et al. 2018, Zheng et al. 2022).

A recent study by Zheng et al. (2022) (hereafter referred to as Z22) opened a new avenue for TIR-based dust retrievals by retrieving ten years of TIR DAOD at 5-km resolution over the global oceans based on combined CALIOP and Infrared Imaging Radiometer (IIR, a collocated higher-spatial-resolution TIR imager) observations. Both CALIOP



and IIR are onboard the CALIPSO satellite. The smaller (compared to AIRS and IASI) footprint size of IIR and the collocated CALIOP lidar make cloud masking much easier and more reliable than stand-alone hyperspectral
algorithms. Moreover, the highly detailed and accurate dust vertical distribution provided by CALIOP not only makes the TIR DAOD retrieval more straightforward and accurate, but also allows for additional retrievals on dust particle size. Lastly, the collocated CALIOP Lidar also provides estimated $DAOD_{VIS}$, which opens potential applications for the observational synergistic VIS and TIR DAOD. Furthermore, unlike the passive VIS-NIR observations that are available at the daytime only, the combined VIS Lidar and TIR observations are also accessible at night, which allows
further applications for investigating the diurnal variability of dust properties (Yu, Y. et al., 2021; Chédin et al., 2020). However, Z22 found that an accurate radiative closure between the simulated TIR radiance and observed TIR radiance for clear sky backgrounds is only possible for nighttime observations as there is an unresolved bias at daytime. In addition, because it used a single-band (i.e., the 10.6 µm IIR band) retrieval method, the algorithm allows for retrieving DAOD only.


To overcome the limitations in Z22 and further advance the TIR dust retrievals for coarse-mode dust size, in this study, instead of IIR, we use three MODIS TIR window bands (centred at 8.55 µm, 11.02 µm and 12.03 µm) for dust retrievals for the following reasons. The detector noise of MODIS in warm scenes (e.g., dust-laden sky) is 0.02-0.03 K, which is lower than that of IIR at 0.1-0.15 K (Madhavan et al., 2016). As a result, we can achieve a better radiative
closure between radiative transfer simulation and MODIS observations in the clear sky, a premise for TIR-based dust retrieval, at all three TIR channels in both daytime and nighttime (details in Section 2.3). It first allows us to adopt the split-window technique (Zhang et al., 2006; Paepe and Dewitte, 2009) to reduce retrieval uncertainties compared with Z22 (detailed in Section 5.1). Moreover, by leveraging the information contents from all three bands, we can retrieve not only DAOD at 11 µm, further scaled to 10 µm (referred to as "$DAOD_{10µm}$"), but also the dust particle size
represented by effective diameter (referred to as "$D_{eff}$"). Lastly, the daytime retrievals enable comparisons with VIS-NIR-based retrievals, such as MODIS, CALIOP and AERONET.

In the rest of the article, we introduce the collocated MODIS and CALIOP observation and the radiative transfer model in section 2. The implementation of the retrieval algorithm is detailed in section 3. Section 4 demonstrates the
$DAOD_{10µm}$ and $D_{eff}$ retrievals of three dust cases observed at Cape Verde and in the Caribbean Sea and compares them with ground-based and in-situ airborne measurements. Section 5 presents the climatological analysis of five-year retrievals of $DAOD_{10µm}$ compared with Z22 IIR-based and IASI-based retrievals and $D_{eff}$ in terms of the seasonal and regional variation from 2013 to 2017. The discussions and conclusions are summarized in section 6.

## 2 Data and model

### 2.1 MODIS and CALIOP observations

In this study, dust properties, namely $DAOD_{10µm}$ and $D_{eff}$, are retrieved from collocated Aqua-MODIS and CALIOP observations. MODIS onboard the Aqua satellite, as a member of the A-train constellation, provides observations from 36 spectral bands ranging from VIS to TIR with near-daily global coverage and relatively high spatial resolution



(i.e., 250 m to 1 km at nadir). MODIS is equipped with onboard calibrators that enable stable calibration uncertainties within ± 0.03 K for TIR bands (Xiong et al., 2009). This study primarily uses the MODIS Level-1B calibrated upwelling radiances at TOA at three TIR spectral bands centered at 8.55 μm, 11.02 μm and 12.03 μm, respectively. The TIR "window" bands mostly avoid contaminations from atmospheric gas absorptions and are sensitive to dust optical properties in different orders (Z22). For better interpretation, the calibrated radiances are further converted to equivalent brightness temperature (BT) computed based on Planck's law, and the corresponding spectral response

functions at the three selected TIR bands (see Figure 4b (black dash lines) in Section 3.2).

CALIPSO, launched in 2006, has also been a member of the A-train constellation that shares a similar and tightly controlled Sun-synchronous polar orbit with Aqua MODIS until August 2018. CALIOP aboard CALIPSO is a two-wavelength (532 and 1064 nm) polarization-sensitive Lidar with three receiver channels (one measuring the 1064 nm

backscatter intensity and two measuring orthogonally polarized components of the 532 nm backscatter). Unlike Aqua MODIS, CALIOP has a much smaller spatial coverage due to its narrow cross-track footprint of around 70 m in diameter. However, the 333-m along-track footprint with 30-to-60-m vertical resolution allows CALIOP to provide detailed vertical structures of aerosols and clouds (Winker et al., 2009).

In this study, the dust contribution to a vertical column of attenuated backscatter is needed as we focus on retrieving dust. Although the CALIOP operational aerosol product (i.e., the Vertical Feature Mask (VFM) product) determines the aerosol sub-type for each aerosol layer (Kim et al., 2018), it does not provide the quantitative dust backscatter profile for dust and non-dust aerosols mixed in the column. Therefore, we apply the estimated particulate depolarization ratio (DPR) profile along with the total attenuated backscatter profile from the Version-4 Level-2

CALIOP aerosol profile product ("LID_L2_05kmAPro-Standard-V4" (Liu et al., 2019)) to derive the dust aerosol's vertical distribution for $DAOD_{10μm}$ and $D_{eff}$ retrieval. The VFM product is further used for filtering out the mis-included non-dust aerosol profiles (see Appendix B for detail).

### 2.2 AMSR-E and MERRA-2 auxiliary data

The surface characteristics (i.e., surface emissivity and temperature) and the atmospheric profiles (i.e., temperature T,

pressure P, water vapor Qv and ozone $O_3$) are crucial for obtaining an accurate radiative transfer simulation in TIR at TOA (Scott and Chedin, 1981). Unlike the hyperspectral observations with the capability to retrieve the instantaneous atmospheric states, our retrieval requires inputs of these auxiliary data from third-party sources.

The atmospheric profiles T, P, Qv and $O_3$, are obtained from Version 2 Modern-Era Retrospective analysis for

Research and Applications (MERRA-2) assimilated products (Gelaro et al., 2017). Specifically, The MERRA-2 "inst3_3d_asm_Nv" product provides 3-hourly instantaneous atmospheric profiles at 72 pressure levels with a gridded horizontal resolution of 0.625° longitude by 0.5° latitude. Detailed information can be found in Gelaro et al. (2017). To assign the gridded MERRA-2 data to the simulations for the collocated MODIS and CALIOP (referred to as



"MODIS-CALIOP") observations, we first obtain the geolocation and time of all grid cells of MERRA-2 data. Then,
we find the spatially and temporally closest grid cell with each MODIS-CALIOP pixel.

As the retrieval is implemented over oceans only, which is explained in Section 2.3, we obtain the level-2 sea surface
temperature (SST) retrieved based on the Advanced Microwave Scanning Radiometer - Earth Observing System
Sensor (AMSR-E) onboard Aqua (ceased operation in December 2011), and its successor (launched in May 2012),
the Advanced Microwave Scanning Radiometer 2 (AMSR2) onboard GCOM-W1 that follows Aqua's orbit. The 6.9-
GHz and 10.7-GHz channels from AMSR-E and AMSR2 are used for SST retrieval (Wentz and Meissner, 2000).
Previous studies demonstrated that the SST retrievals over heavy dust-loading regions using TIR observations are
underestimated due to the radiative impact of dust (Luo et al., 2019). However, microwave radiation has mostly no
interaction with dust and, therefore, can avoid dust impacts and achieve better SST retrieval accuracy over dusty
regions (O'carroll et al., 2019). In this study, the SST at 56 km resolution from AMSR-E and AMSR-2 is collocated
with MODIS-CALIOP. For the surface emissivity, we use the emissivity models (listed in Table 1) provided in
version-2 Community Radiative Transfer Model (CRTM) (Van Delst, 2011), which is described in Section 2.3.

Finally, for each MODIS-CALIOP observation, the collocated MERRA-2 atmospheric profiles, AMSR-E/AMSR2
SST, and the internal surface emissivity model are used as the input for the radiative transfer simulation. All the
satellite products, variables and auxiliary data are listed in Table 1.

Table 1: Values of variables from multi-source satellite sensors and auxiliary datasets that are used in this study

| Satellite sensors | Product names | Variable names | Value is used |
|---|---|---|---|
| MODIS | MYD021KM (CloudSat_MODIS_AUX) | EV_1KM_Emissive | Radiances (BTs) at 8.5 μm, 11μm, 12μm |
| | MYD06 (CloudSat_MOD06_AUX) | Cloud_Phase_Optical_Properties (For daytime) | Clear (0) |
| | | Viewing zenith angle | All |
| CALIOP | LID_L2_05kmAPro-Standard-V4-20 | CAD_score | -100 to -90 |
| | | Particulate_Depolarization_Ratio_Profile_532 | All |
| | | Extinction_QC_Flag_532 | 0,1,16,18 |
| | | Total_Backscatter_Coefficient_532 | All |
| | | Atmospheric_Volume_Description | Three dust subtypes (dust, polluted dust, dusty marine) |
| | CAL_IIR_L2_Track-Standard-V4-20 | Was_Cleared_Flag_1km | No single-shot cloud (0) |
| | | TGeotype | Open Water (1700) |
| AMSR-E AMSR2 | AMSR_E_L2_Ocean RSS_AMSR2_ocean_L3_daily | SST (sea surface temperature) | All |



| Auxiliary data | Product names | Variable names | Value is used |
|---|---|---|---|
| MERRA-2 | Inst3_3d_asm_Nv | H, P, T, QV, O3 | All |
| CRTMv2.3 | Nalli.IRwater.EmisCoeff | Surface emissivity | All |

**2.3 The radiative transfer models**

The foundation of a LUT-based retrieval method is an accurate radiative transfer model. For the radiative transfer simulation of terrestrial TIR radiation under clear atmospheric conditions, atmospheric gaseous absorption is critical. In this study, we use the version-2 CRTM developed by the US Joint Center for satellite data Assimilation (JCSDA) as the foothold for our retrieval (Chen et al., 2012; Han, 2006). The transmittance coefficients in CRTM are first trained by applying regression algorithms to the line-by-line integrated transmittances for numerous atmospheric profiles (McMillin et al., 2006). Afterward, the gaseous absorption component can achieve an accuracy as high as the line-by-line transmittance but consumes far less computational time (Ding et al., 2011). As CRTM also supports MODIS's sensor coefficients, it is an optimal tool for simulating the atmospheric gaseous absorptions at the three selected MODIS TIR bands for our retrieval (Liang et al., 2016; Wang et al., 2016).

Although it is straightforward to use CRTM to handle the gas absorptions in the TIR, we found it difficult to use it to handle the scattering and absorption of dust due to the configuration and structure of the code. In this study, we use the Discrete Ordinate Radiative Transfer code (DISORT) to handle the dust aerosol scattering and absorption calculation (Stamnes et al., 1988). To combine CRTM and DISORT, we first use CRTM to simulate atmospheric gaseous absorptions (output as the atmospheric optical depth) with input MERRA-2 atmospheric profiles. Afterward, the CRTM-simulated atmospheric optical depth with and without the vertical distribution of dust optical properties served as inputs for DISORT to simulate cloud-free dust-laden BTs and cloud-free clean (i.e., cloud-free and aerosol-free) BTs at the three MODIS TIR bands at TOA, respectively.

Prior to implementing the retrieval, the uncertainties contributed by the auxiliary data, the radiative transfer simulation, and the observational errors must be evaluated. Thus, we conduct the radiative closure benchmark between the CRTM-DISORT calculated and the MODIS-observed BTs under cloud-free and clean (without dust) conditions, which is detailly presented in Appendix A. Given that the error of the radiative closure benchmark over land and polar regions can reach up to 10 K due to the uncertainties from the assumed surface emissivity and temperatures (Z22), this study focuses on retrievals over oceans within 60°S and 60°N only.



## 3 Description of the retrieval algorithm

In this section, we detailly describe the retrieval algorithm that is summarized in Figure 1, covering the collocation of MODIS and CALIOP observations and the process of cloud masking and dust detection, the *a priori* dust properties,

and the design of the LUT method and the uncertainty estimation.

**Figure 1: The flow chart of the retrieval process of DAOD$_{10\mu m}$ and D$_{eff}$ using collocated MODIS-CALIOP observations.**



The first step of the retrieval is to identify high-quality cloud-free dust-laden observations. Due to the different spatial
coverage of MODIS and CALIOP, the retrieval requires collocated data from both sensors. The collocation process
and the following cloud masking and dust detection and vertical distribution processing are similar to Z22 and are
presented in detail in Appendix B.

### 3.1 *A priori* dust properties

In addition to the vertical distribution, the retrieval needs to assume dust bulk optical properties. In this section, we
introduce the dust PSD, dust shapes, and dust RI that are used to calculate the bulk optical properties (i.e., the
extinction efficiency ($Q_{ext}$), single scattering albedo (SSA) and asymmetry factor (g-factor)).

### 3.1.1 Monomodal dust coarse-mode particle size distribution

Dust PSD is commonly presented by a two-mode (i.e., fine mode ($D_p < 1.0$ μm) and coarse mode ($D_p > 1.0$ μm))
lognormal size distribution (Dubovik et al., 2002). As the fine mode dust has a negligible effect on TIR observation,
we assume a normalized (i.e., total volume concentration equals unity) monomodal lognormal volume size distribution
to represent the coarse-mode dust PSD, which is defined as

$$\frac{dV}{dlnD} = \frac{1}{\sqrt{2\pi}\sigma} \exp\left[-\frac{ln^2\,(D/D_m)}{2\sigma^2}\right] \qquad (1)$$

where $D$ is the volume-equivalent sphere geometric diameter for spheroidal dust particle assumption (see Section
3.1.2), $D_m$ is the geometric volume median diameter, $\frac{dV}{dlnD}$ is the volume PSD, and $\sigma$ is the standard deviation. Note
that the sensitivity of $\sigma$ to the TIR radiative signature at TOA is negligible compared with that of AOD and $D_m$
(Pierangelo et al., 2005) (see Figure S5). Therefore, to simplify the retrieval, we first set $\sigma = 0.7$ (i.e., ln(2.0)) as it
is a good representation for the coarse-mode dust PSDs in both in-situ measurements and satellite retrievals (Capelle
et al., 2018; Ryder et al., 2018). We further use the effective diameter defined by Hansen and Travis (1974) to represent
the monomodal PSDs with dependence on $D_m$ as

$$D_{eff} = \frac{\int_0^\infty D^3 n(D) dD}{\int_0^\infty D^2 n(D) dD} \qquad (2)$$

where $n(D)$ is the dust number concentration converted by the volume distribution with $\sigma = 0.7$ and varied $D_m$.

Note that in-situ measurements of dust PSD show that the coarsest record of dust particles over the transport regions
(i.e., over oceans) was measured during the Fennec campaign in June 2011 (Ryder et al., 2013a), with an estimated
$D_m$ at around 10.0 μm. Therefore, for the retrieval, we define the minimum and maximum dust coarse-mode PSDs
with their representations of $D_m$ from 1 μm to 12 μm and $D_{eff}$ from 0.8 μm to 9.2 μm, (see Figure S1). Dust PSDs
within this range are used for calculating the dust bulk optical properties as inputs for building the LUT of $DAOD_{10μm}$
and $D_{eff}$.





### 3.1.2 Dust refractive indices and dust shape

The RI of dust, determined by dust mineral compositions, has a profound impact on dust scattering properties and therefore the retrieval results (Sokolik and Toon, 1999). Ideally, the dust RI should be retrieved simultaneously with other properties of dust. However, given the highly limited information content from the three MODIS TIR bands, a retrieval of dust RI is not possible, at least in this study. It should be noted that most previous studies also used pre-assumed dust RI, often one or two simple global constants, in their retrievals including widely used operational aerosol

retrieval products (Capelle et al., 2018; Zhou et al., 2020).

Nevertheless, in this study, we try to incorporate the spatial variability of dust RI in our retrieval by using two newly developed datasets. One is a state-of-the-art dust RI database developed by Di Biagio et al. (2017) (referred to as the "Di-Biagio RI"), which provides dust RIs retrieved based on surface soil samples collected in nineteen arid and semi-

arid sites from worldwide dust source regions. The Di-Biagio RI database provides the observational basis for accounting for the regional dependence of dust RI. The other is the fractional contribution over oceans supplied by various dust source regions from the DustCOMM-2021 dataset developed by Kok et al. (2021a), which is used to assign dust RIs from different source regions to the observed dust aerosol over oceans. Details of the dust RI assignments are presented in Appendix C.


Dust particles have irregular and non-spherical shapes, which vary greatly from case to case and from location to location (Scheuvens and Kandler, 2014). Using spherical assumptions for non-spherical dust in remote sensing would cause significant uncertainty (Huang et al., 2020; Dubovik et al., 2002; Nousiainen and Kandler, 2015). It is essential to adopt a quantified non-sphericity to represent dust optical properties better. However, characterizing the complex

morphology of dust particles remains challenging. Previous studies used different assumptions of dust particle shape to evaluate the sensitivity of dust optical properties to the morphology, such as spheroid (Dubovik et al., 2006), ellipsoid (Meng et al., 2010) and polyhedral (Liu et al., 2013).

The non-sphericity of the aspherical shape is often represented by the aspect ratio, defined as the ratio of the longest

particle dimension to the intermediate particle dimension. The higher the aspect ratio, the greater the non-sphericity. The spheroid shape assumption is a first-order approximation of dust non-sphericity (Mishchenko et al., 1997; Dubovik et al., 2002) and is widely used for non-spherical aerosol retrievals (Levy et al., 2013; Kahn et al., 2010). To seek a broader application of this study to others, we stick to the spheroid assumption with the size-independent aspect ratio distribution from Dubovik et al. (2006) for the retrieval. Nonetheless, the retrieval based on a more advanced

non-spherical dust optical properties database, such as the hexahedral shape (Saito et al., 2021), will be evaluated in future studies.

By assuming dust particles with spheroidal shape, we calculate the dust single-particle optical properties for each *a priori* dust RI using the T-matrix method (Mishchenko et al., 1996). Afterward, the bulk optical properties are

integrated according to the pre-assumed dust PSDs and aspect ratios of spheroidal dust.



### 3.2 The look-up table and the uncertainty estimation

The $DAOD_{10\mu m}$ and $D_{eff}$ are retrieved from three MODIS TIR bands using a LUT method. To illustrate the LUT, we use CRTM-DISORT to simulate the cloud-free clean BT at 11 μm (referred to as "$BT_{11}$"), spectral BT differences (BTD) between 11 μm and 12 μm (referred to as "$BTD_{11-12}$") and that between 8.5 μm and 12 μm (referred to as

"$BTD_{08-12}$"), by giving a typical tropical atmospheric profile with a dust layer distributed at the mid-level troposphere (i.e., 2-6 km, see Figure S6). Afterward, with the *a priori* dust $D_{eff}$, dust RI, and dust spheroidal aspect ratios, the calculated dust bulk optical properties based on the T-matrix method can be used as inputs in CRTM-DISORT to simulate cloud-free dust $BT_{11}$, $BTD_{11-12}$ and $BTD_{08-12}$. With the input DAOD at 11 μm ($DAOD_{11\mu m}$) ranging from 0.0 to 1.0 and $D_{eff}$ ranging from 0.8 μm to 9.2 μm (see Figure S1 for corresponding dust PSDs), we build a LUT consisting

of $BT_{11}$, $BTD_{8-12}$ and $BTD_{11-12}$ as shown in Figure 2a. The assumed dust RI for the LUT is from Algeria in Northeast Africa. Example LUTs corresponding to other dust RIs are shown in Figures S7 and S8.

Considering the higher dust extinction signal expected at 10 μm compared to 11 μm (see Figure 2b and Pierangelo et al. (2004)), we scale $DAOD_{11\mu m}$ to $DAOD_{10\mu m}$ based on the Qe spectral behavior following $\frac{DAOD_{11\mu m}}{DAOD_{10\mu m}} = \frac{Qe_{11\mu m}}{Qe_{10\mu m}}$. As a

result, our final retrieval products contain $DAOD_{10\mu m}$ and $D_{eff}$.

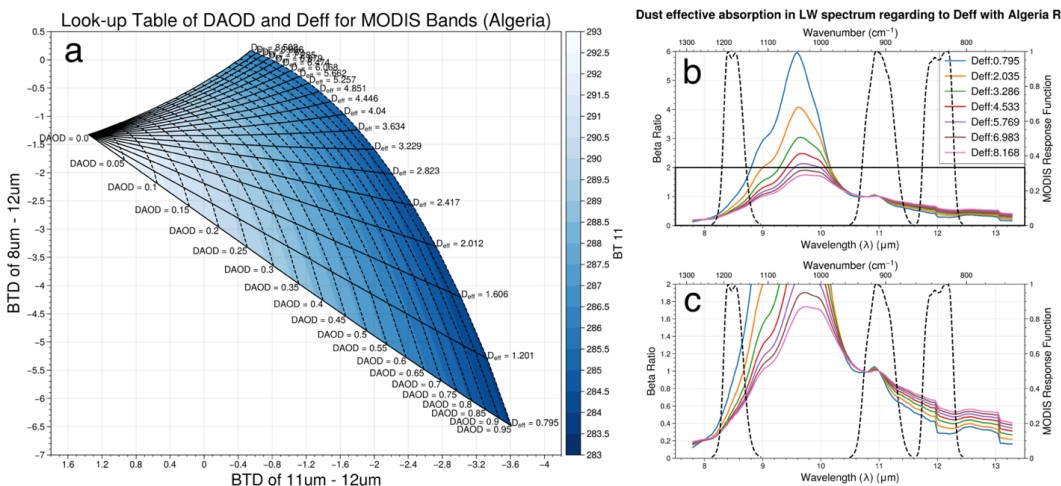

**Figure 2: (a) The example of the LUT of $BTD_{8-12}$ (y-axis), $BTD_{11-12}$ (x-axis) and $BT_{11}$ (colour-filled contours) corresponding to $DAOD_{11\mu m}$ ranging from 0.0 to 1.0 (dashed lines) and $D_{eff}$ ranging from 0.8 μm to 8.2 μm (solid lines) and the Algeria**

**dust RI from Di-Biagio Database. At the point of DAOD = 0.0, the $BTD_{8-12}$ and $BTD_{11-12}$ correspond to the cloud-free clean scenario. (b) The β-ratio to 11 μm calculated based on $D_{eff}$ ranging from 0.8 μm to 8.2 μm and the Algeria dust RI within the TIR spectrum between 7.5 μm and 13.5 μm. (c) The zoom-in area of the black rectangle in (b).**

The variation of BTDs and $BT_{11}$ with $DAOD_{11\mu m}$ and $D_{eff}$ is determined by the dust Qe, SSA and g-factor at the three

TIR bands. To better understand the variations of BTDs in the LUT, we introduce the so-called β-ratio, defined as follows.

$$\beta(\lambda_1/\lambda_2) = \frac{Q_e(\lambda_1)[1-\omega(\lambda_1)g(\lambda_1)]}{Q_e(\lambda_2)[1-\omega(\lambda_2)g(\lambda_2)]} \qquad (3)$$





where $Q_e$ is the extinction efficiency, $\omega$ is the SSA, and $g$ is the asymmetry factor. The $\beta$-ratio was often used to represent the spectral difference of dust "effective absorption" (i.e., absorption and backward scattering) in the TIR spectrum in many studies on dust, volcanic ash and ice cloud retrieval (Pavolonis et al., 2013; Pavolonis et al., 2015; Garnier et al., 2013). Because the variation of $BT_{11}$ in Figure 2a serves as the single-band dust radiative signature, we present the β-ratio with respect to 11 μm (i.e., $\lambda_2 = 11\ \mu m$ in Eq.(3)) for varied $D_{eff}$, as shown in Figure 2b. The β-ratios for wavelength ranging from 12 μm to 11 μm over the whole range of the input $D_{eff}$ are lower than one. It means that the dust "effective absorption" at 11 μm is always more significant than that at 12 μm, regardless of the size variation. In contrast, the cloud-free clean BT at 11 μm is higher than that at 12 μm due to the less atmospheric absorptions at 11 μm as described in Appendix A. Consequently, in Figure 2a, the $BTD_{11-12}$ decreases with increasing $DAOD_{11\mu m}$ regardless of how $D_{eff}$ changes.

On the other hand, the $BTD_{8-12}$ is more sensitive to $D_{eff}$ than to DAOD. First, the cloud-free and clean BT at 8.5 μm is similar to that at 12 μm due to similar gas absorption. However, the dust "effective absorption" at 8.5 μm is larger than that at 12 μm when $D_{eff}$ is relatively small (e.g., 1.0 μm < $D_{eff}$ < 3.0 μm) in Figure 2b, there are negative trends of $BTD_{8-12}$ with increasing DAOD in Figure 2a. In contrast, in Figure 2b, the dust "effective absorption" at 8.5 μm is weaker than that at 12 μm when $D_{eff}$ is relatively large (e.g., $D_{eff}$ > 5.0 μm), leading to positive trends of $BTD_{8-12}$ with increasing DAOD in Figure 2a. In between, the sensitivity of $BTD_{8-12}$ to DAOD can be nearly zero when dust $D_{eff}$ is moderate (e.g., $D_{eff}$ = 3.6 μm). As such, the radiative signature of DAOD and $D_{eff}$ can be separated using $BTD_{8-12}$ and $BTD_{11-12}$, allowing the simultaneous retrieval of both parameters based on the three MODIS TIR bands.

Besides the dust particle size (e.g., $D_{eff}$), dust "effective absorption" at the three TIR bands also depends on the LW dust RI. The dust RI directly changes the spectral behavior of the dust "effective absorption" and reshapes the LUT of $BTD_{8-12}$ and $BTD_{11-12}$ (see Figures S7 and S8). Due to the limited observational signature, the retrieval of dust RI is unachievable in this study. The retrieval uncertainty associated with the assumption of dust RI thus needs to be assessed. In addition, the errors resulting from radiance observation itself and radiative transfer modeling (Figure A1 in Appendix A) also need to be factored in.

We implement the retrieval algorithm in three steps to find an optimal retrieval with the assessed uncertainties. Firstly, we define a cost function ξ of the normalized distance between the simulated BT and BTDs in the LUT and the observed BT and BTDs as

$$\xi(DAOD, Deff) = \frac{1}{3}\left[\frac{\left(BT_{11}^{sim}-BT_{11}^{obs}\right)^2}{\sigma_{11}^2} + \frac{\left(BTD_{11-12}^{sim}-BTD_{11-12}^{obs}\right)^2}{\sigma_{11-12}^2} + \frac{\left(BTD_{8-12}^{sim}-BTD_{8-12}^{obs}\right)^2}{\sigma_{8-12}^2}\right] \quad (4)$$

The subscript of *11,11-12* and *8-12* represents the BT at the 11μm band, $BTD_{11-12}$ and $BTD_{8-12}$, respectively. The superscript of *sim* and *obs* represents the BT or BTD obtained by simulations and observations, respectively. The σ represents the standard deviation of the uncertainty assessed through the clear-sky radiative closure (see Figure A1), which represents the summation of errors from the observation and simulation using *a priori* atmosphere states. The first term on the right-hand-side of Eq. (4) represents the normalized distance between the observed and the simulated



BT at the 11μm band. The second and the last term represents the summation of the normalized distance between the observed and the simulated $BTD_{11-12}$ and $BTD_{8-12}$, respectively.

Secondly, by using Eq. (4), we acquire a solution when the normalized distance is within the range of the evaluated uncertainty ($\xi < 1$). In addition, as mentioned in Appendix C, each observation would possibly assume more than one dust RI for retrieval. Therefore, we build multiple LUTs corresponding to multiple RIs and implement the retrieval with all of them. All the solutions that satisfy $\xi < 1$ in these LUTs are collected.

Finally, the optimal retrieval results of DAOD and $D_{eff}$ are defined as the average of the collected solutions corresponding to multiple *a priori* dust RIs weighted by their corresponding cost function $\xi$ as $w = 1 - \xi$. The weighted standard deviation thus represents the estimated retrieval uncertainty as

$$S_w = \sqrt{\sum_{i=1}^{N} w_i (x_i - \overline{x_w})^2 \Big/ (N-1) \frac{\sum_{i=1}^{N} w_i}{N}} \qquad (5)$$

where $x_i$ is the i[th] solution of DAOD or $D_{eff}$, $w_i$ is the weight of $\xi$ for the i[th] solution, $N$ is the number of non-zero weights, and $\overline{x_w}$ is the weighted mean of the collected solutions (Heckert and Filliben, 2003; Hao and Mendel, 2013). In this step, the uncertainties associated with the assumptions of *a priori* dust RI and the clear-sky radiative closure are taken into account by the weighted average and the weighted standard deviation.

After the retrieval, the quality assurance (QA) flag is assigned as 0 for successfully retrieved results. The retrieval with less than two solutions satisfying $\xi < 1$ is rejected and is assigned with QA flag as 1. By implementing the retrieval for the five-year MODIS-CALIOP observations from 2013 to 2017, which will be analyzed in detail in Section 5, we present the seasonal distribution of cloud-free dust samples ($N_{dust}$), successfully retrieved dust samples ($N_{retrieval}$; QA flag = 0), and the retrieval success rate ($N_{retrieval}$ / $N_{dust}$), which reaches to 90%-100% over dust transport regions, as shown in Figure S9.

In summary (Figure 1), we obtain the cloud-free dust aerosol vertical profiles using the CALIOP cloud mask, dust detection and vertical-scaling method introduced in Appendix B. Afterward, the *a priori* dust properties presented in Section 3.1 serve as inputs for CRTM-DISORT to build the LUT of DAOD and $D_{eff}$. Lastly, we retrieve $DAOD_{11μm}$ further scaled to $DAOD_{10μm}$ and $D_{eff}$ by averaging the solutions that satisfy $\xi < 1$ weighted by $\xi$ and estimate the corresponding retrieval uncertainty based on the corresponding $\xi$-weighted standard deviation.

## 4 Evaluation of CALIOP-MODIS retrievals with in-situ measurements – case studies

### 4.1 A case study for transported Saharan dusts over Cape Verde in summer

In this section, we implement the retrieval to a dust plume originating from North Africa and being transported over the North Atlantic on August 16[th], 2015. We use this case to evaluate the retrieved $DAOD_{10μm}$ and $D_{eff}$ through





comparisons with the in-situ measured dust particle size and the collocated AERONET (Aerosol Robotic NETwork) measurements

**4.1.1 Evaluation of retrieved $DAOD_{10\mu m}$**

Figure 3a shows the total attenuated backscatter at 532 nm from CALIOP for the dust case observed on August 16[th], as shown from left (South) to right (North) with the geolocation highlighted in the upper left sub-panel. The CALIOP orbit passed nearby Cape Verde (16.733°N, 22.935°W) around 03:34 UTC with nighttime observations for the dust plume. Figure 3b shows the corresponding spatial variation of total $AOD_{532nm}$ (blue dots) and $DAOD_{532nm}$ (red dots)

estimated with a Lidar ratio of 44 sr and uncertainty of ±10 sr as described in Appendix B. The mean $DAOD_{532nm}$ (1.1) is ~83% to the mean total $AOD_{532nm}$ (1.33), indicating that "pure" dust aerosols dominate this offshore dust plume. Therefore, as a "golden standard", the measured AERONET AOD at Cape Verde for this dust plume can be approximated as DAOD to assess the CALIOP DAOD and the corresponding retrieved $DAOD_{10\mu m}$. Unfortunately, although AERONET at Cape Verde observed a maximum AOD event at 18:10 on Aug 16[th,] as shown in Figure 3d, it

does not provide any nighttime measurement for this case.

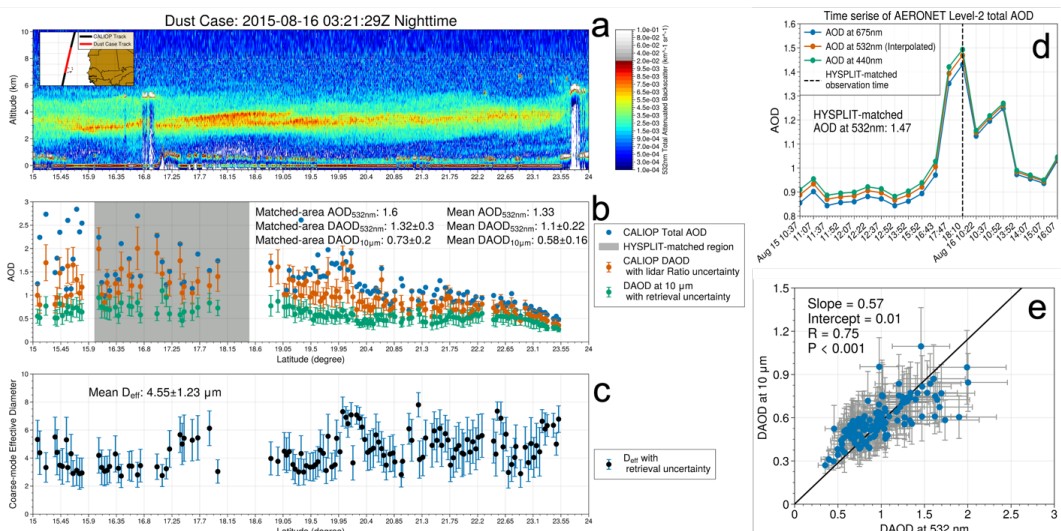

**Figure 3: The nighttime case on August 16[th], 2015. (a) The CALIOP total attenuated backscatter at 532 nm on August 16[th], 2015, over the downwind region of the Sahara Desert (the orbit at upper left). (b) The CALIOP total AOD (blue**

**dots), CALIOP DAOD (red dots), and the retrieved $DAOD_{10\mu m}$ (green dots) of the cloud-free dust-laden profiles. The gray shadow area represents the part of the dust plume (16°N-18°N) observed by CALIOP that is matched with the AERONET measurement based on the HYSPLIT back trajectories as shown in Figure 4 (c) The retrieved $D_{eff}$ (black dots) of the cloud-free dust-laden profiles with the estimated uncertainty (cyan error bars). (d) The time series of the AERONET level-2 AOD at 675 nm (blue dot line), 440 nm (green dot line) and 532 nm (orange dot line, interpolated) at**

**AERONET Cape Verde from 08-15-2015 to 08-16-2015. The black dash lines indicate the time that AERONET measured the same dust plume observed by CALIOP later, proven by the HYSPLIT back trajectories as shown in Figure 4. (e) The**



**scatter plot of DAOD$_{10\mu m}$ versus DAOD$_{532nm}$ for the whole dust case. The grey error bars represent the uncertainties of DAOD$_{10\mu m}$ (vertical) and DAOD$_{532nm}$ (horizontal). The black line represents the robust linear regression with correlation coefficient (R), slope, intercepts, and p-value (P).**

Due to the different observation times and locations between CALIOP and AERONET, to compare their AODs, we present the ensemble back trajectories simulated by the Hybrid Single-Particle Lagrangian Integrated Trajectory (HYSPLIT) model (Stein et al., 2015) from the passing-by times of the MODIS-CALIOP orbits and the AERONET Cape Verde as shown in Figure 4. Note that the vertical distribution of the dust plume is concentrated around 3 km to 4 km (see Figure 3a). Therefore, the HYSPLIT back trajectories are initiated at 3 km and 4 km. In Figure 4a, the HYSPLIT back trajectories of CALIOP between 16°N to 18°N show that the dust plume was seen by AERONET Cape Verde at 18:10 on Aug 15$^{th}$ (see Figure 4b). Bearing in mind that the AOD of the dust plume may change after the the 10-hour transport from AERONET Cape Verde to the CALIOP orbit track, we found that the CALIOP DAOD$_{532nm}$ ($1.32\pm0.3$ averaged from 16°N to 18°N, Figure 3b) is consistent with the AERONET AOD$_{532nm}$ (interpolated, Figure 3d) of 1.47 within its uncertainty.

In addition, both back trajectories from CALIOP (including trajectories > 18°N) and AERONET show similar transport patterns from east to west with initial emission (i.e., back trajectories' height reaches 0 km in Figure 4b) from the source regions (black dash regions in Figure 4a) in Algeria and Mali in both horizontal and vertical view. Thus, we can assign the Di-Biagio RIs from Algeria and Mali as the *a priori* dust RI for retrieval. The retrieved DAOD$_{10\mu m}$ (green dots in Figure 3b) shows a reasonable correlation with DAOD$_{532nm}$ (R = 0.75 in Figure 3e). Because of the spectral difference between TIR and VIS, the HYSPLIT-matched mean DAOD$_{10\mu m}$ (0.73) is ~55% of the value of DAOD$_{532nm}$ (1.32). For the entire case, DAOD$_{10\mu m}$ is ~57% (k = 0.57 in Figure 3e) of DAOD$_{532nm}$. Note that both ratios are within the empirical range of the TIR-to-VIS DAOD ratio from 28% to 65% (Peyridieu et al., 2013), depending on the assumptions of dust PSD, dust RI and dust non-spherical shape.





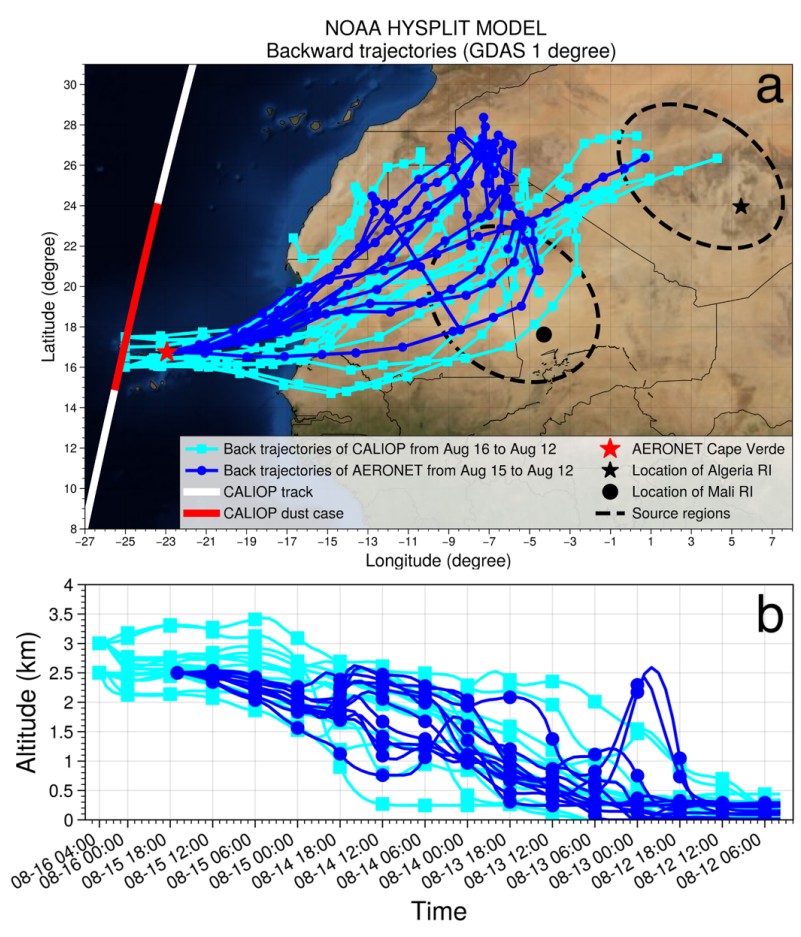

**Figure 4: The distribution of the spatial (a) and the vertical (b) ensemble HYSPLIT back trajectories on the CALIOP dust case on 08-16-2015 (cyan rectangle solid lines) and the AERONET Cape Verde observation for the dust case on 08-15-2015 (blue dot solid lines). The black star and circle represent the geolocation of Di-Biagio RI collected over Algeria**
**and Mali, respectively. The red star denotes the geolocation of the AERONET Cape Verde site.**

### 4.1.2 Comparison of $D_{eff}$ with AER-D in-situ measurements

In this section, we evaluate the $D_{eff}$ retrieval by comparing the $D_{eff}$-corresponding monomodal PSD with those deduced from the lognormal-fitted dust PSD measured during the AERosol Properties – Dust (AER-D) campaign from Aug
$7^{th}$ to Aug $25^{th}$, 2015, over the outflow region of North Africa around Cape Verde (Ryder et al., 2018). We also compared with the AERONET-version-3 Level-2 two-mode PSD (referred to as "AERONET PSD") on Aug $16^{th}$ at Cape Verde (Dubovik et al., 2006) (see Figure 5b). The AER-D campaign provides measured dust PSD and the corresponding uncertainty for dust within the Sahara Air Layer (SAL) (see Figure 5a) and in the marine boundary layer from various airborne instruments (Ryder et al., 2018). In this case, as the CALIOP-observed dust plume is well



confined within 2-5 km (Figure 3a), we choose the AER-D SAL campaign mean log-fit size distribution (referred to as "AER-D PSD"), which is measured within 1.2-4.8 km (Ryder et al., 2018), in our comparison.

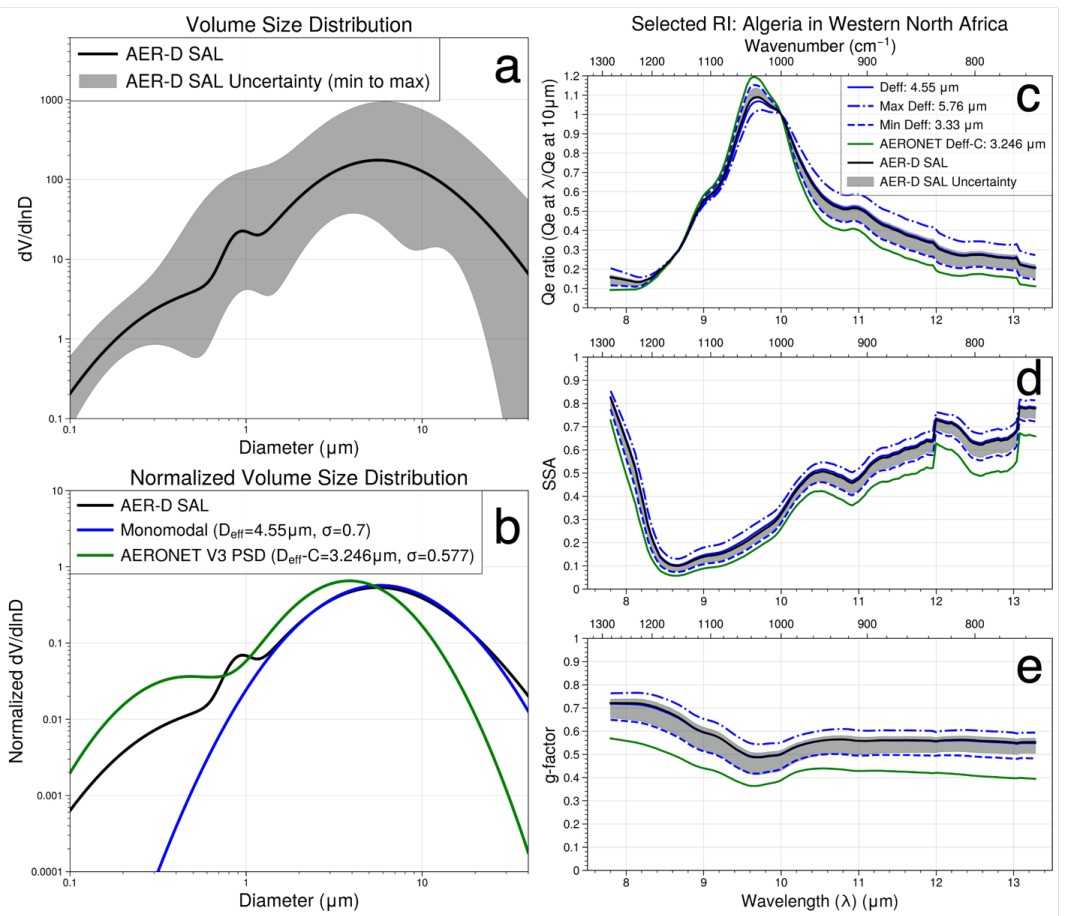

**Figure 5: (a) The volume lognormal (*dVdlnD*) AER-D PSD (black) with gray shadow area indicating the min-to-max range**
**of the measurement uncertainty obtained from Ryder et al. (2018). (b) The normalized *dVdlnD* of AER-D (black), the**
**retrieved coarse mode PSD corresponding to $D_{eff}$ = 4.55 μm (blue) and the AERONET PSD (green). (c-e) The Qe ratio to**
**10 μm (c), SSA (d), g-factor (e) calculated based on the AER-D PSD (black) with min-to-max uncertainty (grey shadow),**
**retrieved coarse mode PSD (blue) with the retrieval uncertainty (blue dash curve for the lower bound; blue dash-dot curve**
**for the upper bound) and AERONET PSD (green).**

Figure 3c shows the retrieved $D_{eff}$ for the dust case. We found that the spatial variation of $D_{eff}$ is generally positively correlated with that of $DAOD_{10μm}$ and with a mean value of 4.55 μm with the uncertainty ranging from 3.33 μm to 5.76 μm. Therefore, we obtain the monomodal PSD corresponding to $D_{eff}$ = 4.55 μm to compare with the normalized AER-D PSD and the AERONET PSD. In Figure 5b, the AEROENT-retrieved coarse mode PSD is systematically





smaller than that of AER-D, while the monomodal PSD with $D_{eff}$ = 4.55 μm agrees well with the AER-D coarse mode
         PSD, although the fine mode of AER-D PSD is not compared because not relevant for LW.

         For a perhaps more relevant comparison of the three PSDs in coarse mode, we compare their corresponding optical
         properties, namely the dust Qe ratio at 10μm, SSA and g-factor in the TIR spectrum ranging from 8 μm to 13 μm (See
Figures 5c to 5e). The reason for using the dust Qe ratio at 10 μm is that the retrieved $DAOD_{10μm}$ provides a constraint
         of dust extinction at 10 μm, while the dust PSD further determines the spectral Qe ratio of other TIR wavelength to
         10 μm. We found that the spectral Qe ratio, SSA and g-factor calculated based on the monomodal PSD used in our
         retrievals are consistent with that calculated based on AER-D PSD. It demonstrates that although our monomodal PSD
         lacks fine mode dust, the retrieved $D_{eff}$ can still provide almost identical dust optical properties in TIR as the AER-D
PSD has based on the constraint from the retrieved $DAOD_{10μm}$. In other words, the combination of $DAOD_{10μm}$ and
         $D_{eff}$ with comparable accuracy as in-situ measurements but better spatiotemporal coverage is a valuable tool for
         reducing the global mean LW dust DRE uncertainties due to DAOD and dust particle size.

         On the other side, all three optical properties calculated based on the AERONET PSD are bias low compared with
that based on AER-D and the retrieved PSD. As the fine-mode PSD has a negligible impact on dust optical properties
         in TIR, the result suggests that the AERONET coarse-mode PSD is highly likely to be underestimated in terms of
         size, which has been pointed out in several studies comparing AERONET PSD with other in-situ measurements
         (Müller et al., 2010; Müller et al., 2012; Mcconnell et al., 2008; Adebiyi et al., 2023). Due to the difficulties of
         comparing the PSD from the column-integrated retrieval to that from the lofted-layer measurement (Toledano et al.,
2019), the possible reasons are as-of-yet not well-explained, which require detail investigations in the future.

### 4.2 Evaluation of $D_{eff}$ with SAMUM-2 and SALTRACE in-situ measurements

Noting that limiting the validation of $D_{eff}$ with one case may be biased. To better demonstrate the reliability of the $D_{eff}$
retrieval, we compare the retrieved $D_{eff}$ with in-situ measured dust PSDs from two additional field campaigns. One is
from the second field experiment of the Saharan Mineral Dust Experiment project (SAMUM-2) in the Cape Verde
area during January to February 2008 (Weinzierl et al., 2011). The other is the Saharan Aerosol Long-Range Transport
         and Aerosol–Cloud-Interaction Experiment (SALTRACE) that is taken place over the North Africa, the Atlantic
         Ocean, and the Caribbean from June to July 2013 (Weinzierl et al., 2016).

### 4.2.1 Comparison of $D_{eff}$ with SAMUM-2 – A case over Cape Verde in winter

To compare with the SAMUM-2 campaign, we perform the retrieval for a nighttime dust case observed eastward Cape
Verde on Jan 28, 2008, as shown in Figure 6. Figure 6a shows the dust case is a low-altitude-level case up to ~ 2 km,
         consistent with the dust sampling of the experimental flights on January 28[th] in SAMUM-2 (see Table 2 and Figure 9
         in Weinzierl et al. (2011)). As shown in Figure 6b, the mean retrieved $D_{eff}$ is 3.68 μm, which is further used to construct
         the corresponding PSD for the comparison with SAMUM-2 dust PSD (see Figure 6c). In this case, we implement the
         retrieval using the RI assignments introduced in Section 3.1.2 and Appendix C instead of performing HYSPLIT back



trajectories to identify dust source regions. In addition, this wintertime dust case has lower dust loading (not shown)
than the summertime case in Section 4.1, leading to a lower information content for retrieving $D_{eff}$ (see Figure 2a).
Therefore, there is a mean retrieval uncertainty of 2.0 μm larger than the summertime case.

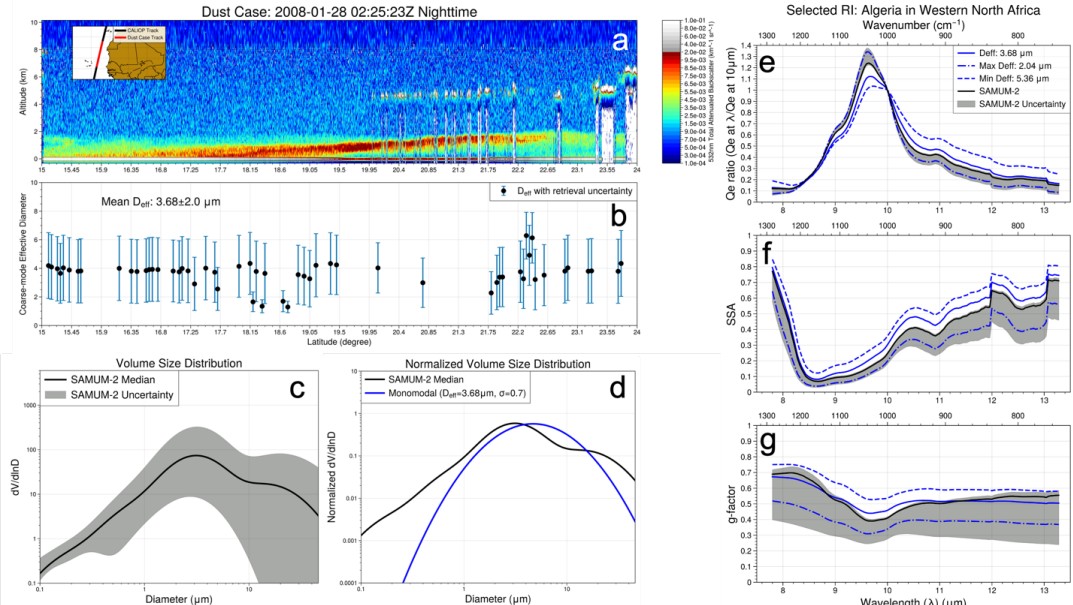

**Figure 6: (a) The CALIOP total attenuated backscatter at 532 nm on January 28[th], 2008, over the downwind region of the**
**Sahara Desert (the orbit at upper left). (b) The retrieved $D_{eff}$ (black dots) of the cloud-free dust-laden profiles with the**
**retrieval uncertainty (cyan error bars). (c) The volume lognormal ($dVdlnD$) SAMUM-2 PSD (black) with gray shadow area**
**indicating the range of the measurement value from 3- to 97- percentile obtained from Weinzierl et al. (2011). (d) The**
**normalized $dVdlnD$ of SAMUM-2 (black) and the retrieved coarse mode PSD corresponding to $D_{eff}$ = 3.68 μm (blue). (e-g)**
**The Qe ratio to 10 μm (e), SSA (f), g-factor (g) calculated based on the SAMUM-2 PSD (black) with its uncertainty (grey**
**area) and the retrieved coarse mode PSD (blue) with the retrieval uncertainty (blue dash curve for the lower bound; blue**
**dash-dot curve for upper bound).**

In Figure 6d, we found that the monomodal PSD corresponding to the mean retrieved $D_{eff}$ agrees with SAMUM-2
PSD by having the peak between the third and fourth modes of SAMUM-2 PSD. Due to the limitation of the fixed
assumption of the lognormal volume distribution's standard deviation, the monomodal PSD overestimates dust with
$D_p$ from 4 to 13 μm but underestimates dust with $D_p > 13$ μm. Because of that, the Qe ratio, SSA and g-factor
corresponding to the monomodal PSD have slight differences from that of the SAMUM-2 PSD in the TIR spectral
region. However, the dust TIR optical properties of the two PSDs are generally consistent after considering their
uncertainties. It shows the $D_{eff}$ retrieval's capability to capture the seasonal differences of dust size in the Cape Verde
area revealed by the AER-D and SAMUM-2 filed campaigns.



**4.2.2 Comparison of $D_{eff}$ with SALTRACE – A dust case transport throughout North Atlantic from June 12[th] to June 23[rd], 2013**

In order to evaluate the $D_{eff}$ retrieval at long-range transport regions and demonstrate the variation of $D_{eff}$ during the transport, we compare our results with the dust $D_{eff}$ measured during the SALTRACE field experiment that studied a

Lagrangian dust plume over both Cape Verde (SALTRACE-E) and Barbados (SALTRACE-W) on June 17[th] and June 22[rd], 2013.

First of all, we perform the retrieval on a series of MODIS-CALIOP observations from June 16[th] within the Cape Verde area to June 23[rd] over the Caribbean Sea, as shown in Figure 7. In Figure 7a, the dust plume was vertically

distributed between 2 km to 6 km, with the mean retrieved $D_{eff}$ at 4.8 μm (Figure 7a2) on June 16[th]. From June 18[th] to 20[th], the dust plume was transported to the mid-Atlantic (~43°W) and decreased the layer height from 2-5 km to 2-4 km (Figures 7b1 and 9c1). Meanwhile, the mean retrieved $D_{eff}$ reduced from 4.3 μm to 4.0 μm (Figures 7b2 and 7c2). Figures 7d1 to 7f1 show that the dust plume traveled toward the Caribbean Sea from June 21[st] to June 23[rd], maintaining the layer height between 1.5 km and 3.5 km and the retrieved $D_{eff}$ at ~3.9 μm (Figures 7d2 to 7f2). During the transport,

the dust loading is also decreasing (see Figures 7a1 to 7f1), leading to lower information content for retrieving $D_{eff}$ (see Figure 2a) and, therefore, relatively higher retrieval uncertainty (error bars in Figures 7a2 to 7f2).

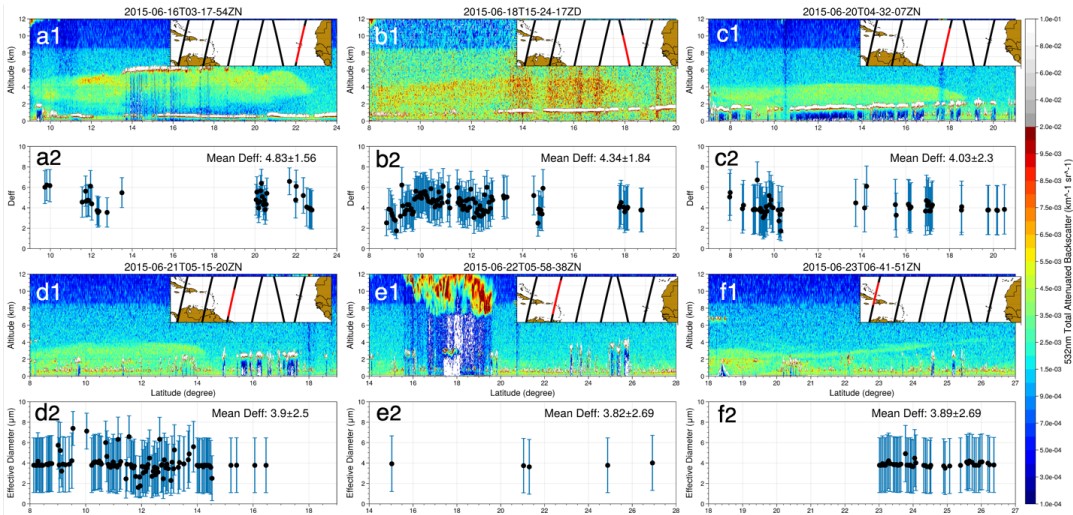

**Figure 7: (a1 to f1) The CALIOP total attenuated backscatter at 532 nm on June 16[th] (a1), 18[th] (b1), 20[th] (c1), 21[st] (d1), 22[nd] (e1), 23[rd] (f1), 2013 (the orbit at upper left). (a2 to f2) The retrieved $D_{eff}$ (black dots) of the cloud-free dust-laden profiles**

**with the retrieval uncertainty (cyan error bars) corresponding to a1 to f1.**

To prove that the MODIS-CALIOP observations snapshotted the transport processes of the same dust case that SALTRACE observes, we present the HYSPLIT back trajectories started from the MODIS-CALIOP observation on June 23[rd] over the Caribbean Sea, as shown in Figure 8. We set the dust layer heights at 2 km and 3.5 km at the starting

point (see Figure 8b) to serve as the vertical boundaries of the observed dust plume on June 23[rd] (Figure 7f1). Figure



8a shows that the dust event originated from the North African source regions identified during SALTRACE (see Figure 5 in Weinzierl et al. (2016)) before June 13[th]. In addition, the dust case has transport trajectories overlapping with the MODIS-CALIOP observed dust plumes presented in Figure 7. Comparing Figure 8b with Figures 7a1 to 7f1, we found that the vertical height of the dust plume varies between 2 km and 6 km during transport, which agrees with

the vertical dust distribution observed by MODIS-CALIOP. Therefore, we conclude that the MODIS-CALIOP dust cases observe the same dust case as SALTRACE. However, we notice that the MODIS-CALIOP observational times are not perfectly consistent with the back trajectory times, implying that the retrievals presented in Figure 7 may not be the properties of the same air mass in the dust event as observed by SALTRACE. Thus, we do not expect a perfect agreement between our retrieved $D_{eff}$ and the SALTRACE measurements.


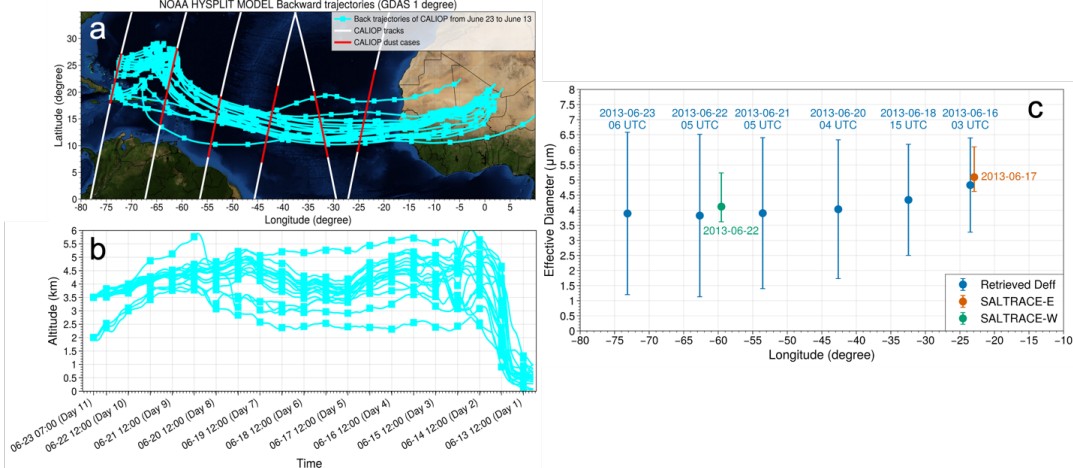

**Figure 8: The distribution of the spatial (a) and the vertical (b) ensemble HYSPLIT back trajectories on the CALIOP dust case from 2013-06-23 back to 2013-06-12 (cyan rectangle solid lines). The white solid curves represent the MODIS-CALIOP orbit tracks that observed the dust cases presented in Figure 7, which are highlighted in red curves. (c) The $D_{eff}$**

**versus the longitudes of the MODIS-CALIOP retrievals in Figure 7 (blue dots), the SALTRACE-E at Cape Verde (red dot) and the SALTRACE-W at Barbados (green dot). The corresponding error bars represent their retrieval uncertainties and in-situ measured uncertainties.**

Figure 8c shows the retrieved mean $D_{eff}$ of the MODIS-CALIOP observed dust plumes in Figure 7 and the $D_{eff}$ of

SALTRACE-E at Cape Verde and SALTRACE-W at Barbados. The retrieved $D_{eff}$ on June 16[th] over Cape Verde (4.8 µm) is close to that of SALTRACE-E (5.1 µm). During the transport from Cape Verde (23°W) to mid-Atlantic (43°W), the $D_{eff}$ decreases from 4.8 µm to 4.0 µm. Approaching Barbados and further the Caribbean Sea, the $D_{eff}$ remains at 3.9 µm, which is also close to that of SALTRACE-W (4.1 µm). It validates the $D_{eff}$ retrieval in both the short-range and long-range transport regions and demonstrates the retrieval's capability of revealing the transport process of dust

coarse-mode particle size in a better spatiotemporal resolution than in-situ measurements.



## 5 Climatological analyses

### 5.1 Comparison of $DAOD_{10\mu m}$ with IIR-based $DAOD_{10.6\mu m}$ and IASI-based $DAOD_{10\mu m}$

Beyond the case studies, we statistically evaluate the MODIS-CALIOP $DAOD_{10\mu m}$ by comparing it with the three independent TIR-based satellite-retrieved $DAOD_{TIR}$ datasets that are rigorously assessed through comparisons with
AERONET $COD_{500nm}$. The first one is the night-time-only IIR-based $DAOD_{10.6\mu m}$ from Z22. Note that the IIR-based retrieval has two $DAOD_{10.6\mu m}$ datasets based on two different dust PSD assumptions. We use the one based on the Fennec SAL PSD from Ryder et al. (2013a) (referred to as "Fennec-SAL $DAOD_{10.6\mu m}$") recommended from Z22. The second one is the IASI-based dataset, as mentioned in Section 1. It retrieves $DAOD_{10\mu m}$ and dust mean layer altitude based on a two-step LUT method developed by the research group at Laboratoire de Météorologie Dynamique (LMD)
(referred to as "IASI-LMD"). The third one retrieves $DAOD_{10\mu m}$ using IASI based on an artificial neural network (NN) method developed by the research group at Université libre de Bruxelles (ULB) (referred to as "IASI-ULB") (Clarisse et al., 2019).

As the IIR Fennec-SAL $DAOD_{10.6\mu m}$ can be easily collocated with the MODIS-CALIOP $DAOD_{10\mu m}$, we perform a
pixel-by-pixel comparison between the two datasets using the five-year retrievals from 2013 to 2017 at nighttime based on a two-step collocation method. As mentioned in Z22, the IIR-based retrieval is implemented on samples with the estimated $DAOD_{532nm} > 0.05$, while the MODIS-CALIOP retrieval does not carry this limitation. Consequently, we first choose the MODIS-CALIOP $DAOD_{10\mu m}$ with the corresponding estimated $DAOD_{532nm} > 0.05$ in both products. Note that the MODIS-CALIOP $DAOD_{10\mu m}$ is retrieved simultaneously with $D_{eff}$ while the IIR Fennec-SAL
$DAOD_{10.6\mu m}$ was retrieved with a fixed $D_{eff} \sim 6.7$ μm. Therefore, to control the dust PSD impact on the retrieved DAOD, we further select the MODIS-CALIOP $DAOD_{10\mu m}$ with $D_{eff}$ ranging from 4 μm to 8 μm to collocate with the Fennec $DAOD_{10.6\mu m}$.

As shown in Figure 9a, the $DAOD_{10\mu m}$ correlates with $DAOD_{10.6\mu m}$ with R = 0.7 and with $DAOD_{10.6\mu m}$ being
systematically lower than $DAOD_{10\mu m}$ by 25% (slope = 0.75). The difference may be attributed to the spectral difference between 10.0 μm and 10.6 μm (see Figure 5c), which ranges from 0.5 to 0.8 for $D_{eff}$ ranging from 4 μm to 8 μm. In addition to spectral differences, several factors may have caused the variability of the collocated pixels between the two datasets. Firstly, although the collocated $DAOD_{10\mu m}$ are pre-selected based on $D_{eff}$. The impact of dust PSD on the retrieved DAOD still exists, as it is challenging to find enough pixels with the same $D_{eff}$ as the Fennec SAL
observation. Secondly, the treatments of RI in the two studies are also different. In Z22 a relatively simple method is used to assign the Di-Biagio RI to different regions while we utilize the DustCOMM-2021 to help assign Di-Biagio RI in this study (see Appendix C). Furthermore, the MODIS-CALIOP retrieval has evolved from the IIR-based retrieval from Z22 with three improvements, namely the lower detector noise from MODIS, the improved retrieval methods, and the enhanced dust RI assumptions. These differences may have also directly affected the pixel-by-pixel
comparison.




Thanks to the abovementioned improvements, in Figure 9b, the histogram of the absolute DAOD uncertainty of $DAOD_{10\mu m}$ is reduced by ~55% from 0.2 to 0.09 in terms of the mean value compared with that of $DAOD_{10.6\mu m}$. In Figure 9c, the relative uncertainty of $DAOD_{10\mu m}$ is substantially reduced compared with that of $DAOD_{10.6\mu m}$, especially

for retrievals with small DAOD value (e.g., $DAOD_{10\mu m} < 0.1$). Consequently, we conclude that the MODIS-CALIOP $DAOD_{10\mu m}$ is generally consistent with the IIR-based $DAOD_{10.6\mu m}$ from Z22 with a substantial improvement regarding the retrieval uncertainty

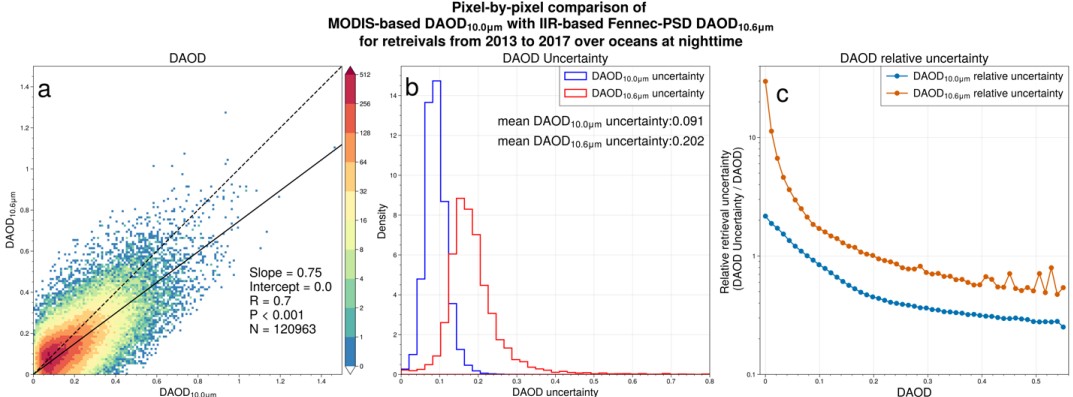


**Figure 9: The pixel-by-pixel comparison of MODIS-CALIOP DAOD_{10μm} with the IIR-based Fennec-SAL DAOD_{10.6μm} from Z22 for retrievals from 2013 to 2017 over oceans at nighttime. (a) The joint histogram of DAOD_{10μm} and DAOD_{10.6μm}. The solid black line is the linear regression of the two datasets. The R, P and N at the lower right represent the Pearson correlation coefficient, p-value and the number of pixels of the linear regression. (b) The probability density**

**function (PDF) of DAOD_{10μm} uncertainty (blue) and DAOD_{10.6μm} uncertainty (red). (c) The mean relative retrieval uncertainty (i.e., DAOD uncertainty / DAOD) of DAOD_{10μm} (blue) and DAOD_{10.6μm} (red). The y-axis is on a logarithmic scale.**

Unlike the comparison with the IIR-based retrieval, the orbit difference between the MODIS-CALIOP and IASI

observations and the cloud-free dust sampling difference between the corresponding retrievals prevent the pixel-by-pixel comparison with the Level-2 data (Zheng et al., 2022). Therefore, we alternatively perform the climatological comparison among the aggregated 5° longitude by 2° latitude Level-3 seasonal mean MODIS-CALIOP, IASI-LMD and IASI-ULB $DAOD_{10\mu m}$ based on five-year data from 2013 to 2017 in both daytime and nighttime. The 5° by 2° seasonal mean IASI-LMD and IASI-ULB $DAOD_{10\mu m}$ are aggregated from the corresponding 1° by 1° monthly mean

Level-3 products. Note that both seasonal mean IASI DAODs are divided by the total number of AOD samples. To be consistent, in our retrieval, the seasonal mean $DAOD_{10\mu m}$ is averaged by the total number of cloud-free aerosol samples ($N_{aerosol}$ in Figure S9). Similar to Z22, we highlight three dust-transport regions, North Atlantic (NA), Indian Ocean (IO) and North Pacific (NP) as shown in Figure 10.

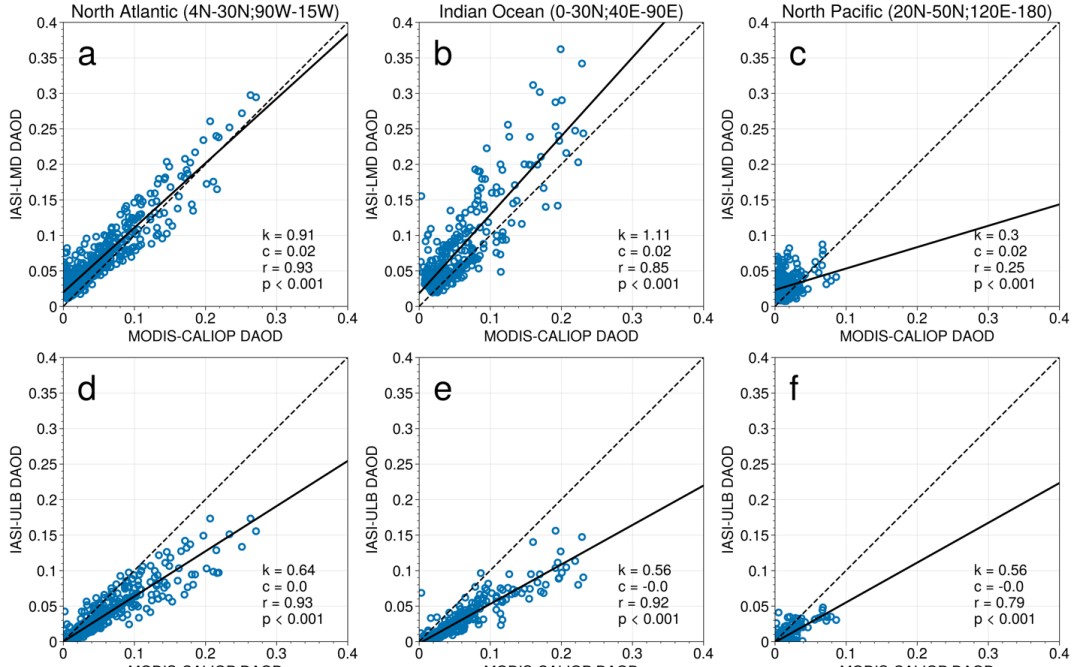

**Figure 10: The comparisons of the seasonal mean MODIS-CALIOP DAOD$_{10\mu m}$ versus IASI-LMD (a,b,c) and IASI-ULB (d,e,f) DAOD$_{10\mu m}$ over NA (a,d), IO (b,e) and NP (c,f) from 2013 to 2017. Each point in the scatterplots represents a seasonal mean DAOD$_{10\mu m}$ in one of the 5° by 2° grids for a specific year from 2013 to 2015. The solid black lines are the linear regressions of each comparison, while the black dash lines are the reference one-to-one lines. The k, c, r and p at the upper right of each panel represent each linear regression's slope, intercept, correlation coefficient and p-value.**

Figure 10 shows the comparisons of 5° by 2° gridded seasonal mean MODIS-CALIOP DAOD$_{10\mu m}$ with the two IASI DAOD$_{10\mu m}$ over the three dust-transport regions from 2013 to 2017. MODIS-CALIOP DAOD$_{10\mu m}$ over NA and IO are highly correlated and consistent with IASI-LMD DAOD$_{10\mu m}$ with R = 0.9,0.8 and k = 0.9,1.1, respectively, as shown in Figures 10a and 10b. However, for optically thin dust (e.g., DAOD < 0.1), IASI-LMD DAOD$_{10\mu m}$ is systematically ~0.02 greater than MODIS-CALIOP DAOD$_{10\mu m}$ as shown from the intercepts of the linear regression in Figures 10a and 10b. In addition, the two datasets have poor agreement over NP (R = 0.2). In contrast, MODIS-CALIOP DAOD$_{10\mu m}$ achieves a better correlation over the dust-transport regions (including NP) than the comparison with IASI-LMD DAOD$_{10\mu m}$ (see Figures 10d-10f). It is mainly because both retrievals mostly avoid contamination from sub-pixel clouds and background aerosols, which should be the reason for the high bias of the IASI-LMD optically thin dust (DAOD$_{10\mu m}$ < 0.1) DAOD$_{10\mu m}$. However, IASI-ULB DAOD$_{10\mu m}$ is 40%-60% lower than that of MODIS-CALIOP and IASI-LMD DAOD$_{10\mu m}$ over the three dust-transport regions (k = 0.6,0.5,0.4 in Figures 10d-10f).





Although the discrepancy of the two IASI-based $DAOD_{10\mu m}$ is non-negligible, the two datasets achieve good
agreements with AERONET because the different assumptions of TIR-to-VIS DAOD ratios offset the $DAOD_{10\mu m}$
difference. As they assumed similar dust RIs (e.g., OPAC RIs) and spherical dust, the disagreement of the $DAOD_{10\mu m}$
could be due to the different *a priori* dust PSDs. IASI-ULB assumes a monomodal dust PSD with a geometric mean
radius at 0.5 μm (i.e., mean diameter at 1.0 μm). It is much smaller than the IASI-LMD assumed coarse-mode dust
PSD with an effective radius at 2.3 μm (i.e., effective diameter = 4.6 μm), possibly leading to the systematically lower
$DAOD_{10\mu m}$ compared with MODIS-CALIOP and IASI-LMD $DAOD_{10\mu m}$. Because MODIS-CALIOP $D_{eff}$ has a
climatological value ranging from 4.0 to 5.0 (see Figure 11, detail in Section 5.2), which is closer to 4.6 μm, MODIS-
CALIOP $DAOD_{10\mu m}$ shows better consistent with the IASI-LMD $DAOD_{10\mu m}$. This non-negligible dependency of the
retrieved $DAOD_{10\mu m}$ to dust PSD is also presented in Z22. It highlights the importance of the spatiotemporal variation
of dust PSD and the advantage of the observational constraint on both $DAOD_{10\mu m}$ and dust coarse-mode PSD from
the MODIS-CALIOP retrieval.

### 5.2 Spatiotemporal variation of dust $D_{eff}$

One of the main objectives of this study is to provide a climatological view of dust coarse-mode size variation during
transport in a global coverage from satellite observations, which has not yet been available in the literature, as far as
we know, due to the difficulties mentioned in Section 1. In this section, we present the spatiotemporal variation of $D_{eff}$
in terms of seasonal variation, the regional difference among dust-transport regions and longitudinal-mean variation
within dust-transport regions based on the five-year retrieval data from 2013 to 2017.

Different with seasonal mean $DAOD_{10\mu m}$, the denominator of the seasonal mean $D_{eff}$ is the number of samples with
successful retrievals only. Noting that the retrieval uncertainty is large for optically thin dust (i.e., $DAOD_{10\mu m} < 0.1$,
see Figure 9c), we consider that the seasonal mean $DAOD_{10\mu m} < 0.005$ is mainly contributed by optically thin dust and
therefore mask the seasonal mean $D_{eff}$ with seasonal mean $DAOD_{10\mu m} < 0.005$ to focus on more confident $D_{eff}$
retrievals. We found that the seasonal variation of $D_{eff}$ is highly correlated with that of $DAOD_{10\mu m}$. For example, the
largest $D_{eff}$ over NA and IO occurs in summer, while the peak of $D_{eff}$ over NP happens in spring. As dust extinction in
TIR is more sensitive to coarse-mode dust (Ryder et al., 2019), it is reasonable to find that the greater the $DAOD_{10\mu m}$,
the coarser dust is in the atmosphere.

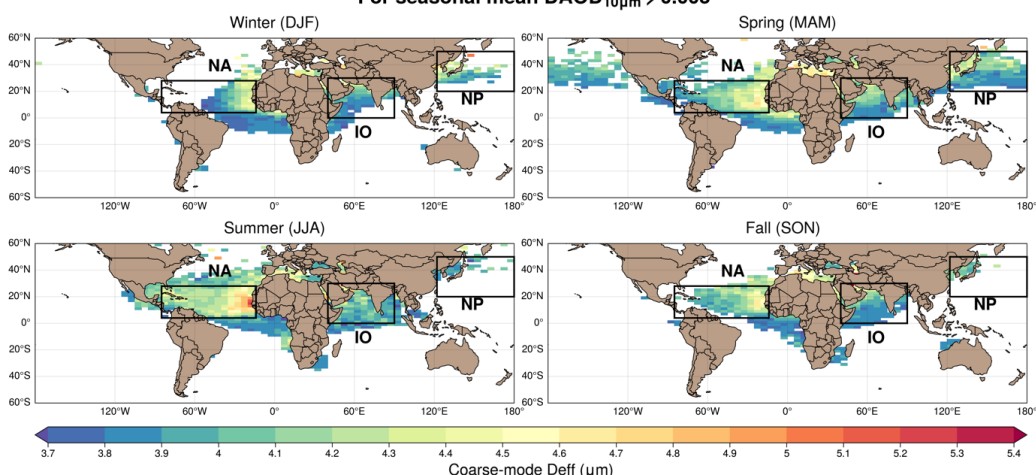

**Figure 11: 2013-2017 five-year averaged seasonal mean D$_{eff}$ masked by the five-year seasonal mean DAOD$_{10\mu m}$ > 0.005 in winter (a), spring (b), summer (c) and fall (d). The black boxes indicate the three defined dust-transport regions as same as**
**in Figure 9.**

In terms of regional differences, we found that the maximal seasonal D$_{eff}$ over IO (~4.2 μm in summer, Figure 11c) and NP (~4.2 μm in spring, Figure 11b) are ~22% lower than that over NA (~5.4 μm in summer, Figure 11c). It suggests that the coarse-mode dust is frequently found over NA but not over IO and NP. It is expected over NP because

the transport distance from source regions located in East Asia to NP is much longer than that from North Africa to NA (Alizadeh-Choobari et al., 2014). In addition, the emitted and transported dust PSD from Asia is possible to be finer than that from North Africa and Arabia due to the radiative feedback on dust emission (Woodage and Woodward, 2014). As a result, there is less chance for coarse dust particles to survive till over NP. In contrast, although dust from the Middle East to IO has a similar transport distance as that over NA, fewer coarse dust particles are found over IO.

As the retrieval samples are distributed similarly over the three transport regions for all seasons as shown in Figure S9, it is less likely to have sampling bias between regions. One of the possible reasons that the long-range transport dust from the Middle East is not within elevated mixed layers (Carlson, 2016), such as SAL over NA, that triggers static instability and strong vertical turbulence to sustain coarse dust particles for a longer lifetime (Gutleben and Groß, 2021; Gasteiger et al., 2017). However, due to insufficient in-situ measurements on dust PSD in the Middle East

(Adebiyi et al., 2020b), what causes the regional differences in dust particle size after long-range transport remains open and needs further investigation in the future.

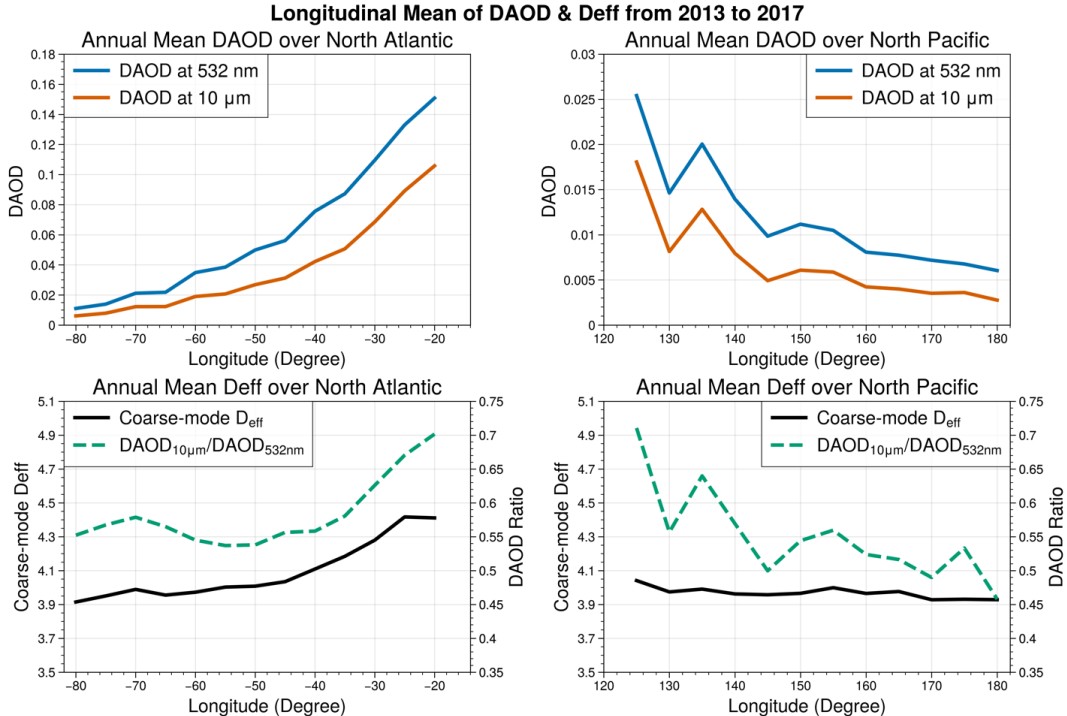

**Figure 12: The annual longitudinal mean DAOD$_{10\mu m}$ (blue curves), DAOD$_{532nm}$ (red curves), the DAOD ratio of DAOD$_{10\mu m}$ to DAOD$_{532nm}$ (green dash curves) and D$_{eff}$ (black curves) over the North Atlantic (a,c) and the North Pacific (b,d).**

As mentioned in Section 5.1, the assumed dust PSD impacts heavily on both the retrieved DAOD$_{10\mu m}$ and the theoretical TIR-to-VIS DAOD ratio. Because our retrieval provides simultaneous DAOD$_{10\mu m}$ and D$_{eff}$ and the synergetic CALIOP estimated DAOD$_{532nm}$, we further investigate the relationship between D$_{eff}$ and the observational-based TIR-to-VIS DAOD ratio (defined as DAOD$_{10\mu m}$/DAOD$_{532nm}$). From the view of longitudinal transport for Saharan dust and Asian dust, we demonstrate the annual longitudinal mean DAOD$_{532nm}$ and DAOD$_{10\mu m}$ over NA and NP (Figures 12a and 12b). They both show a consistent variation between DAOD$_{532nm}$ and DAOD$_{10\mu m}$ with a decreasing trend westward over NA and eastward over NP, convincing the reliability of the DAOD$_{10\mu m}$ retrievals described in Section 5.1. Figures 12c show that the corresponding DAOD ratio over NA decreases by 23% westward from 18°W to 40°W and 8% from 40°W to 80°W. As coarse-mode particles dominate the dust extinction in TIR, the reduction of the DAOD ratio reflects the faster decrease of DAOD$_{10\mu m}$ and, thus, implies a loss of coarse-mode dust loading in the column, which also can be inferred by the decreasing of D$_{eff}$. In Figure 12c, we found that the decreasing trends of D$_{eff}$ and DAOD ratio are highly correlated. It demonstrates a ~20% reduction of DAOD ratio and a ~7% decrease of mean D$_{eff}$ during the transport to the mid-Atlantic, while there are less than 10% and 2% decrease of DAOD ratio and mean D$_{eff}$ during the rest of the NA transport. The transport pattern over NA is also similar with the dust case observed by SALTRACE presented in Section 4.2.2. However, in Figure 12d, we found a rapid fluctuation of the DAOD ratio due to the fewer retrieval samples (Figure S9) and higher retrieval uncertainty from 120°E to



140°E, which is 'gainst $D_{eff}$'s relatively stable decreasing rate. Despite that, we found a less than 10% reduction of $D_{eff}$ throughout the eastward transport to 180°, suggesting a stable trend of dust coarse-mode size during the NP transport.


Note that dust particle size varies in the day-to-day transports, which is not visible in the long-term averaged longitudinal transport. To provide details on the variation of $D_{eff}$ in different size ranges during transport over NA and NP in their peak season, we present the population distribution of $D_{eff}$ longitudinally in summer over NA and spring over NP. We first slice the NA region from 4°N to 30°N into seven sub-regions at 10° longitude intervals, as shown

in Figure 13h. Within each sub-region box, we present the histogram and the cumulative distribution function (CDF) of $D_{eff}$ of all the optically thick dust (i.e., $DAOD_{10\mu m} > 0.1$) in the summer from 2013 to 2017. To better visualize the variation of $D_{eff}$ during the transport, within each sub-region box, the histogram and CDF (blue curves in Figures 13a to 13g) are compared with those from the previous (i.e., eastward) box are also presented (red curves in Figures 13b to 13g).

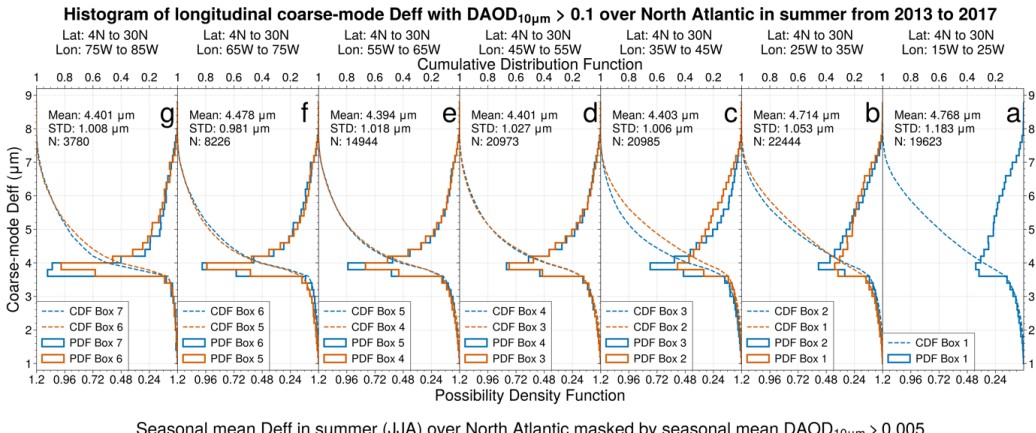

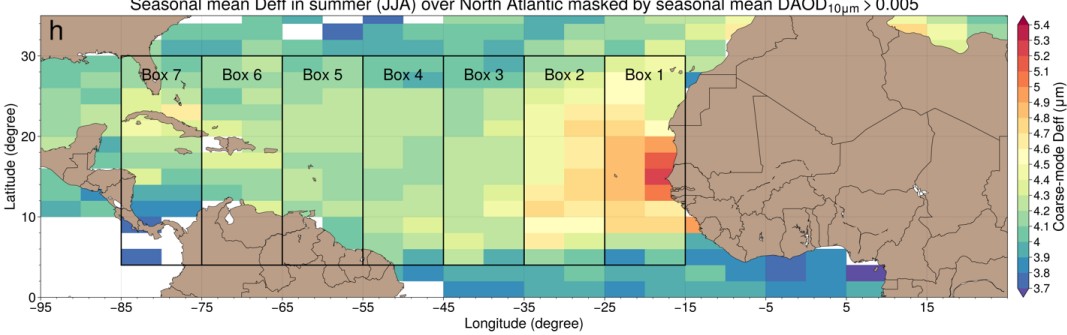


**Figure 13: The histogram (solid curves with the bottom x-axis) and the cumulative distribution function (dash curves with the top x-axis) of $D_{eff}$ with $DAOD_{10\mu m} > 0.1$ within each longitudinal box from west to east ranging from 1 (a) to 7 (g) over the NA in the summer from 2013 to 2017. Blue curves represent $D_{eff}$ samples within the current box. Orange curves represent $D_{eff}$ samples within the previous box eastward. (h) The geolocation boundaries of each longitudinal box on top of**
**the seasonal mean $D_{eff}$ were masked by seasonal mean $DAOD_{10\mu m} > 0.005$ over the NA in the summer from 2013 to 2017.**




Figures 13a to 13b show a slightly decrease of the population of $D_{eff}$ greater than 5.5 μm. From Figures 13b to 13c, according to the CDFs, the contribution of $D_{eff} > 5$ μm to the total number reduced from 40% to 20%. In PDFs, there is a ~50% reduction in the population of $D_{eff} > 5$ μm, while ~20% more dust with $D_{eff} \sim 4$ μm is found, leading to the

reduced mean $D_{eff}$ from 4.7 μm to 4.4 μm. Meanwhile, the peak of the PDFs in Figures 13a to 13g remains stable at 4.0 μm, while the number of samples decreases gradually, as shown in PDFs in Figures 13d to 13g, indicating that less coarse-mode dust can be transported to Boxes 5 to 7 (55°W to 85°W).

The result suggests that ~50% of relatively coarser dust ($D_{eff} > 5$ μm) tends to drop out when transported to the mid-
Atlantic (25°W to 35°W), which is ~2000 km away from source regions over North Africa. Afterward, from the mid-Atlantic to the Caribbean Sea, the mean $D_{eff}$ remains almost unchanged, agreeing with the arguments from previous studies that the stabilization of coarse-mode dust PSD during the long-range transport (Weinzierl et al., 2017; Denjean et al., 2016; Ryder et al., 2019). Additionally, dust samples with $D_{eff} > 5$ μm can still be found even at 65°W and 75°W (see Figures 13f and 13g) but with a relatively lower frequency (~20%). In other words, super-coarse dust particles,
although rare, are still possible to be carried on a long-distance journey during the transport over NA (Van Der Does et al., 2018), which act against the gravitational settling theory by Stoke's law (Ginoux, 2003; Bagnold, 1974).

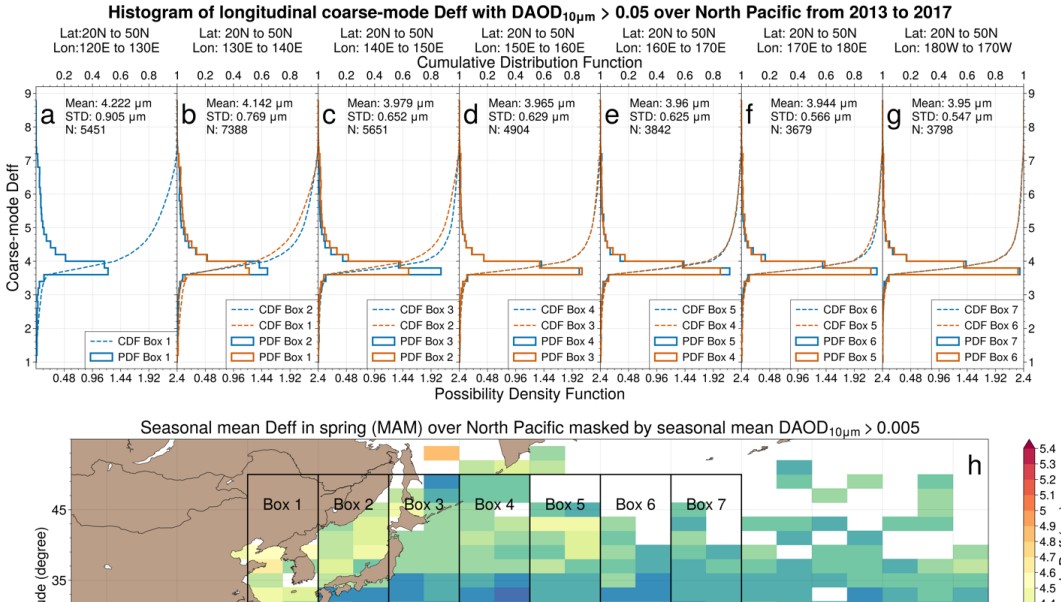

**Figure 14: Same as Figure 13 but for $D_{eff}$ with $DAOD_{10μm} > 0.1$ within each longitudinal box from west to east ranging from 1 (a) to 7 (g) over the NP in the spring from 2013 to 2017.**






Unlike the Trans-Atlantic dust, Asian dust transport over NP experiences longer travel distances from the East Asian source regions and therefore has systematically smaller particle size. With the same method as Figure 13, we present the histograms of $D_{eff}$ with $DAOD_{10\mu m} > 0.05$ (according to the lower seasonal mean $DAOD_{10\mu m}$ over NP) within seven longitudinal boxes, as shown in Figure 14. During the transport, the mostly unchanged PDFs with $D_{eff} > 4$ μm from

Box 1 (130°E) to Box 2 (140°E) only reduced ~10% from Box 2 to Box 3 (150°E). The CDFs are stable throughout the transport from Box 3 to Box 7 regardless of the total number of dust samples decrease. Compared with dust over NA, only ~5% of dust with $D_{eff} > 5$ μm can be found after transporting to Box 3. The relatively homogenous and stable distribution of $D_{eff} \sim 4$ μm suggests that the coarse-mode dust particles over NP may also have longer lifetimes than expected by the stand-alone gravitational settling theory. However, as $DAOD_{10\mu m}$ and the number of successful

retrievals over NP are lower than that over NA and IO, the relatively higher retrieval uncertainty prevents us from drawing a clear conclusion. Future studies are recommended to validate the satellite retrieved $D_{eff}$ by in-situ measured Asian dust PSDs.

**6 Discussions and Conclusions**

This study developed a novel retrieval algorithm for $DAOD_{10\mu m}$ and the coarse-mode dust PSD represented by $D_{eff}$ using the collocated CALIOP and MODIS observations. The $D_{eff}$ retrieval is detailly validated through three case studies in Aug 2015, Jan 2008 and June 2013, respectively.

We validate the $DAOD_{532nm}$ matching with the AERONET total AOD at Cape Verde in the 2015 case study. Despite

the spectral difference preventing the "apple-to-apple" comparison of $DAOD_{10\mu m}$ with AERONET, the relatively good correlation between $DAOD_{10\mu m}$ and the AERONET-validated $DAOD_{532nm}$ demonstrate the $DAOD_{10\mu m}$ retrieval's reliability. Afterward, we present the consistency of the monomodal PSDs corresponding to the retrieved $D_{eff}$ with the AER-D PSD and SAMUM-2 PSD as well as their TIR optical properties in the 2015 and 2008 cases. The 2013 case validates the $D_{eff}$ retrieval in both the short-range (Cape Verde) and long-range (the Caribbean Sea) transport regions

by comparing with SALTRACE dust PSD and demonstrates the retrieval's capability of revealing the transport process of dust coarse-mode particle size in a better spatiotemporal resolution than in-situ measurements. The results convince us that the $DAOD_{10\mu m}$ and $D_{eff}$ retrieval dataset can provide a better constraint on regional and global LW DRE uncertainties due to DAOD and dust PSD. However, an assumption of dust RI is still needed.

We apply the retrieval to five-year MODIS-CALIOP data from 2013 to 2017 and compare the DAOD retrieval with IIR-based and IASI-based retrieval. As an improved version compared with the IIR-based retrieval, the MODIS-CALIOP retrieval reduced ~50% of DAOD uncertainty and achieves good consistency (R = 0.7 in Figure 9). In the climatological comparison with the seasonal mean IASI-based $DAOD_{10\mu m}$, MODIS-CALIOP $DAOD_{10\mu m}$ reaches a better agreement with IASI-LMD $DAOD_{10\mu m}$ over NA (R = 0.9) and IO (R = 0.8) than that over NP (R = 0.2).

Meanwhile, the IASI-ULB $DAOD_{10\mu m}$ over the three regions are highly correlated with MODIS-CALIOP $DAOD_{10\mu m}$, while they are systematically underestimated possibly due to the $D_{eff}$ of the pre-assumed dust PSD is significantly



lower than that of IASI-LMD $DAOD_{10\mu m}$ and the climatological $D_{eff}$. The discrepancy of the two AERONET-evaluated IASI $DAOD_{10\mu m}$ datasets reveals that the dependency of TIR-to-VIS DAOD ratios and the retrieved $DAOD_{10\mu m}$ to dust PSD is non-negligible, which is also proved in Z22. It highlights the importance of considering the spatiotemporal variation of dust $D_{eff}$ in TIR retrievals.

A global and climatological analysis of the five-year $D_{eff}$ retrievals from -60°N to 60°N over oceans is presented. Comparing $D_{eff}$ among the three transport regions, we found that seasonal mean $D_{eff}$ over IO (3.9-4.2 μm) is up to ~22% lower than that over NA (4.1-5.4 μm) depending on different seasons, implying a shorter lifetime of coarse-mode dust particles transported from the Middle East to IO than that from North Africa to NA. For $D_{eff}$ variation during transport, over NA, we found a ~50% reduction of retrievals with $D_{eff}$ > 5 μm from 15°W to 40°W and a relatively stable $D_{eff}$ at ~ 4 μm throughout the Caribbean Sea. The $D_{eff}$ result from 15°W to 40°W differs from the IASI-retrieved effective radius distribution over NA in Peyridieu et al. (2013), which presented an almost constant value at 2 μm during summer throughout the transport. In addition, the prevailing dust with $D_{eff}$ at ~ 4 μm and a small portion of dust with $D_{eff}$ > 5 μm (5%-20%) found after long-term transport in both NA and NP can hardly be explained by the stand-alone gravity settling theory. The results provide observation-based transport patterns of the coarse mode dust size over oceans, which can be used to evaluate the simulated dust coarse mode PSD in dust transport models.

However, there are serval limitations for our retrieval. First of all, the case-study validation of $D_{eff}$ is limited to three field campaigns. Extended comparisons with other in-situ measurements, especially over Mediterranean, IO and NP, should be realized further to validate the applicability and significance of the proposed approach. Secondly, our retrieval is still not applicable for observations over land due to the uncertainties from land surface temperature and emissivity. Nonetheless, with more reliable databases of land surface characteristics, this portable retrieval algorithm can be easily extended to cover the dust source regions. Thirdly, the limited spatial coverage of CALIOP restrains the application of our data to regional studies. Extending the retrieval to off-CALIOP-track MODIS pixels is recommended for future studies.





**Appendix A: The cloud-free clean radiative closure benchmark between the CRTM-DISORT calculated and**
**the MODIS-observed BTs**

In this study, the uncertainties contributed by the auxiliary data, the radiative transfer simulation, and the observational errors is evaluated through the radiative closure benchmark between the CRTM-DISORT calculated and the MODIS-observed BTs under cloud-free and clean (without dust) conditions based on the collocated MODIS and CALIOP data from 2007 to 2010.


Figures A1a and A1c show the BT discrepancies (referred to as "dBT") between the simulations and the observations for daytime and nighttime cloud-free and clean cases over oceans at three MODIS TIR bands. Figures A1b and A1d show the corresponding discrepancies of the cloud-free spectral BT differences (BTD) between 11 μm and 12 μm (blue curve, referred to as "$dBTD_{11-12}$") and that between 8.5 μm and 12 μm (red curve, referred to as "$dBTD_{08-12}$").
Both dBTs and dBTDs are unbiased (i.e., with a peak and a mean value centered at zero) in both daytime and nighttime, demonstrating a remarkable consistency between the CRTM-DISORT simulation and the MODIS observation. For the three single TIR BTs, because the 8.5-μm and 11-μm bands are the cleaner (i.e., less water vapor absorption) than the 12-μm band that is the most water-vapor-absorptive, the standard deviations of dBT at 8.5 μm and 11 μm are lower (0.78 – 0.86 K) than that at 12 μm (0.88 – 0.9 K). Interestingly, the errors of the two dBTDs are substantially reduced
to the range of 0.2 to 0.4 K as the errors from the assumed atmospheric states at each band are canceled out, especially the $dBTD_{11-12}$ (0.18 K at nighttime and 0.21 K at daytime), which is sensitive to DAOD as explained in section 3.2. The smaller uncertainties in dBTDs are probably due to error cancellations. For example, if AMSR-E underestimates the SST, the simulated BT would be colder than the observation because of the overestimated surface-emitted radiance. However, the underestimation happens in all three TIR bands, and the error cancels each other to some
extent, leading to smaller uncertainty in dBTDs.

Overall, one standard deviation of dBTs and dBTDs represents the retrieval uncertainty due to the atmospheric auxiliary data, the radiative transfer simulation, and the observational errors, which is revisited in section 3.2.



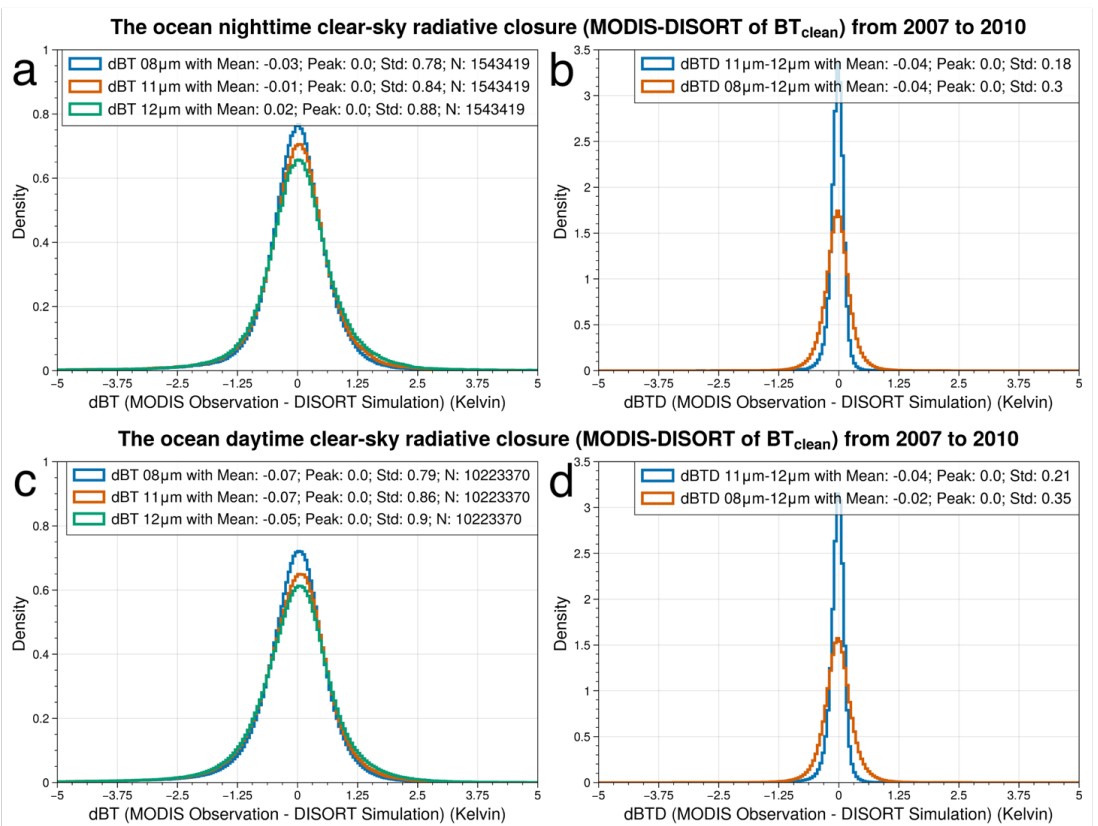

**Figure A1: The nighttime (a) and daytime (c) cloud-free and clean sky dBTs at the MODIS 8.5-µm (blue), 11-µm (orange) and 12-µm (green) bands. The nighttime (b) and daytime (d) cloud-free clean sky dBTD$_{8-12}$ (orange) and dBTD$_{11-12}$ (blue).**

### Appendix B: Pre-processing of the cloud-free dust detection from the collocated MODIS and CALIOP observation

The first step of the retrieval is to identify high-quality cloud-free dust-laden observations. Due to the different spatial coverage of MODIS and CALIOP, the retrieval requires collocated data from both sensors, which is done in the following steps. First, we refer to the MODIS-AUX product (Partain, 2007) developed for CloudSat to find along-CloudSat-track MODIS pixels for two reasons. First, each along-CloudSat-track profile has 15 collocated MODIS pixels in the MODIS-AUX product. Each MODIS pixel contains the MODIS Level-1B radiances and Level-2

geometries. Using the 15 collocated MODIS pixels saves computational time from accessing the original terabyte-scale Level-1B products (Zheng et al., 2021). Second, the along-track orbits of CALIPSO and CloudSat are highly synchronized. It allows each along-CALIPSO-track profile to quickly match the nearest pixel among the 15 collocated along-CloudSat-track MODIS pixels. However, noting that the MODIS viewing zenith angle of the collocated pixels are not exactly nedir as CALIOP has, which is also considered in our retrieval (See Table 1).





The CALIOP "LID_L2_05kmAPro-Standard-V4" product has a 5 km along-track resolution, while the MODIS-AUX
product is 1 km. To address this spatial difference, we reference the 1-km along-CALIOP-track geolocation records
from the "IIR_L2_Track-Standard-V4" product, which provides five records with 1-km resolution in each 5-km
CALIOP profile. Each of the five 1-km geolocation records is then used to find the nearest along-CloudSat-track

MODIS pixels. The corresponding MODIS Level-1B 1-km TIR BTs, BT uncertainties and the MODIS sensor's
geometries (i.e., viewing/solar zenith/azimuth angles) are then assigned to each geolocation record.

Similar to the cloud masking process as in Z22, we use the collocated 1-km "Was_Cleared_Flag_1km" originated
from the "IIR_L2_Track-Standard-V4" product to screen out MODIS pixels containing sub-pixel clouds that are

detected by the single-laser-shot in 333 m along-track footprint. Finally, the remained cloud-free 1-km MODIS pixels
within each 5-km CALIOP profile are averaged, forming the 5-km collocated MODIS-CALIOP cloud-free product.

After cloud masking, dust detection also follows the procedures described in Z22. Firstly, we identify the high-quality
CALIOP backscatter profiles by applying the extinction control flag ("Extinction_QC_Flag_532 = 0, 1, 16, 18"

(Winker et al., 2013; Yu et al., 2015a)) to the 5-km MODIS-CALIOP cloud-free product. Next, we apply the cloud-
aerosol discrimination (CAD) score to select the profiles containing all detected features with CAD between -100 to
-90 to ensure the quality of the detected aerosol layers (Yu et al., 2019). Finally, the selected CALIOP backscatter
profiles are further used to distinguish dust from non-dust aerosols. The separation is based on the contrast of the DPR
between dust and non-dust aerosols. The higher the non-sphericity and particle size, such as dust, the lower the DPR.

Therefore, the DPR of dust aerosols ($\delta_d$) is usually higher than that of other non-dust aerosols ($\delta_{nd}$). Accordingly, a
vertically resolved fraction $f_d(z)$ of dust backscatter ($\beta_d(z)$) to the observed backscatter ($\beta(z)$) (i.e., $f_d(z) = \beta_d(z)/\beta(z)$), is estimated by the observed particulate DPR $\delta(z)$, $\delta_d$ and $\delta_{nd}$ as

$$f_d(z) = \frac{(\delta(z)-\delta_{nd})(1+\delta_d)}{(1+\delta(z))(\delta_d-\delta_{nd})} \quad (B1)$$

Following Yu et al. (2015a) and Z22, the lower and upper limits of $\delta_{nd}$ are set to 0.02 and 0.07 and $\delta_d$ to 0.20 and

0.30, respectively. The final $f_d(z)$ is set to the mean value of the upper bounds ($\delta_d = 0.3$ and $\delta_{nd} = 0.07$) and lower
bounds ($\delta_d = 0.2$ and $\delta_{nd} = 0.02$). Due to the observed particulate DPR uncertainty, the value of $f_d(z)$ can exceed 1
or below 0, which are set to be 1 and 0, respectively. Finally, we obtain the backscatter profile of dust aerosol as

$$\beta_d(z) = f_d(z) \cdot \beta(z) \quad (B2)$$

which serves as the dust vertical distribution to scale the input DAOD in the CRTM-DISORT simulation. Note that

the extinction coefficient profile can be obtained by multiplying $\beta_d(z)$ with an *a priori* dust extinction-to-backscatter
ratios (i.e., Lidar ratios (LR)) for dust aerosol. According to previous studies for the dust LR (Haarig et al., 2022; Liu
et al., 2002; Liu et al., 2008; Kim et al., 2020), we further calculate the column integrated DAOD at 532 nm (referred
to as "DAOD$_{532nm}$") by assuming a dust LR at 44 sr with $\pm10$ sr uncertainty. However, the DAOD$_{532nm}$ uncertainty
contributed by LR is beyond this study and will not be discussed. Readers are referred to Kim et al. (2020) for details.






However, the DPR-based method is likely to include the contribution of sea salt over open oceans and generate non-zero $DAOD_{532nm}$ even without dust, especially in daytime when CALIOP has lower quality due to the solar contamination. The possible reason is that sea salt would have DPR close to dust when their relative humidity is low (e.g., < 50%) (Haarig et al., 2017). Therefore, after deriving $DAOD_{532nm}$, we further use the CALIOP VFM (i.e., the 920 "Atmospheric_Volume_Description" in the "LID_L2_05kmAPro-Standard-V4" product; see Section 2.1 and Table 1) to filter the profiles that has no dust, or polluted dust or marine dust layers. Finally, the rest of profiles are considered as cloud-free dust profiles for retrievals.

**Appendix C: Assignments of dust long-wave refractive index**

Note that assigning dust RIs from different source regions to the observed dust aerosol over the ocean should follow the dust global transport patterns. Accordingly, we applied the fractional contribution over oceans supplied by various dust source regions from the DustCOMM-2021 dataset developed by Kok et al. (2021a). This dataset provides the seasonally resolved global distribution of the fractional contribution of DAOD from nine defined dust source regions (see Figure C1) by integrating observational constraints on dust properties and abundance into an ensemble of GCM 930 simulations, gridded with a resolution of 2.5° longitude by 1.9° latitude. In other words, for each grid cell in each season, there are nine fractions representing the contributions from nine source regions, indicating the probability of where the DAOD that occurs in a grid cell in a particular season originated. We use this data set to choose the appropriate *a priori* dust RIs from different source regions for our retrieval over oceans.

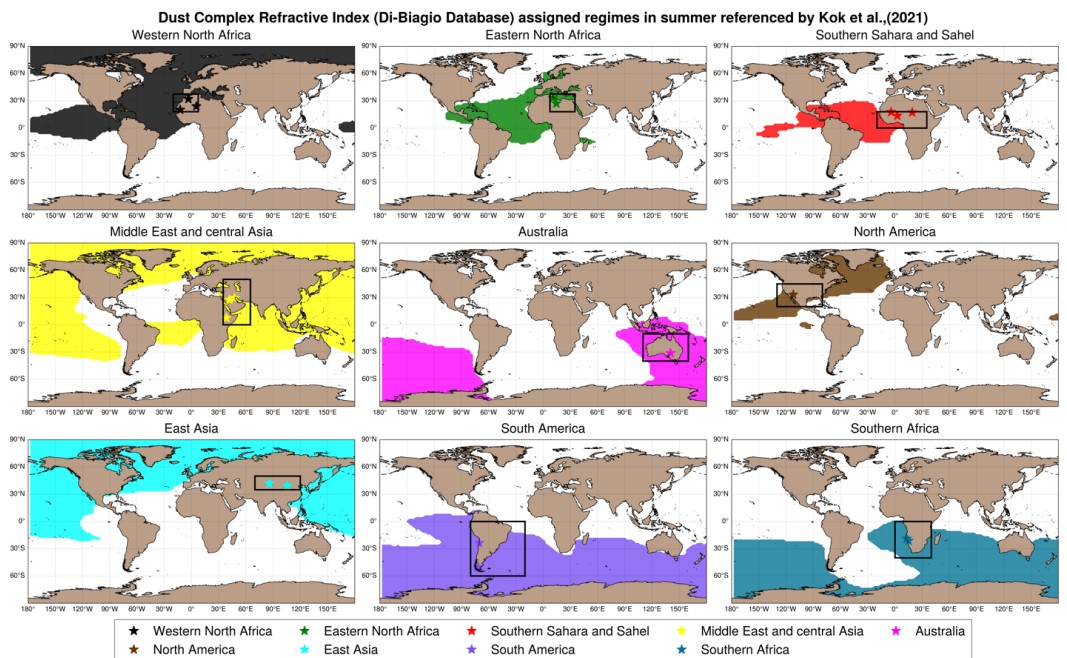



**Figure C1: The assignment of the source region-resolved dust refractive indices from Di Biagio et al. (2017) is based on which of the nine main source regions provided a fractional contribution to SW DAOD that exceeds 0.1, which is shown here for summer based on the DustCOMM-2021 dataset.**

It should be noted that the DustCOMM-2021 is a climatological dataset. As such the uncertainties included in this
dataset cannot be propagated into not propagable for instantaneous observational retrievals. Hence, in this study, instead of scaling the Di-Biagio RI's fractional contribution to form a new dust RI, we select the source region if its DAOD fractional contribution exceeds 0.1 and assign the corresponding Di-Biagio RIs within the selected source region for our retrieval. Figure C1 shows the nine regimes over oceans with fractional contribution greater than 0.1 for the nine defined dust source regions in summer (see Figures S2 to S4 for other seasons). The Di-Biagio RIs are
assigned to the nine source regions based on their geolocations. For observations in each season within each regime, the retrieval will assume the dust originated from the identified dust source regions and choose the corresponding Di-Biagio RIs. Note that the nine regimes can overlap, meaning that the observation over a particular grid cell covered by multiple regimes will assume multiple RIs from these regimes. The uncertainty due to the variation of multiple RIs is evaluated in Section 3.2.

**Data availability**

The 2007 to 2017 MODIS-CALIOP $DAOD_{10\mu m}$ and coarse-mode $D_{eff}$ data (L2 and $5° \times 2°$ monthly L3) will be publicly available after the acceptance of this manuscript.

**Author contribution**

Conceptualization, J.Z. and Z.Z.; methodology, J.Z., Z.Z. and Y.H.; software, J.Z., Z.Z., Q.S. and C.W.; validation,
J.Z. and Z.Z.; formal analysis, J.Z.; investigation, J.Z., and Z.Z.; data curation, J.Z., Z.Z., J.F.K., C.D.B., C.R.; writing—original draft preparation, J.Z.; writing—review and editing, J.Z., Z.Z., Y.H., A.G., Q.S., C.W., J.F.K., C.D.B., C.R. and Y.D.; visualization, J.Z.; supervision, Z.Z., Y.H. and A.G.; project administration, Z.Z.; funding acquisition, Z.Z. All authors have read and agreed to the published version of the manuscript.

**Declaration of Competing Interest**

Zhibo Zhang reports that financial support was provided by NASA.

**Acknowledgment**

JZ, ZZ and AG are supported by a NASA grant (no. 80NSSC20K0130) from the CALIPSO and CloudSat program managed by David Considine. HY was supported by the CloudSat/CALIPSO program. CR acknowledges funding from NERC grant NE/M018288/1. JFK was supported by NSF grants 1856389 and 2151093 and the Army Research
Office (cooperative agreement number W911NF-20-2-0150). C.D.B. was supported by the Centre National des Etudes





Spatiales (CNES) and by the CNRS via the Labex L–IPSL, which is funded by the ANR (grant no. ANR–10–LABX–0018). The computations in this study were performed at the UMBC High Performance Computing Facility (HPCF). The facility is supported by the US National Science Foundation through the MRI program (grant nos. CNS-0821258 and CNS-1228778) and the SCREMS program (grant no. DMS-0821311), with substantial support from UMBC. We acknowledge the AERIS data infrastructure for providing access to the IASI-LMD data in this study and CNRS-LMD for the development of the retrieval algorithms. We thank the ICARE Data and Services Center for providing access to the IASI-ULB data in this study at http://www.icare.univ-lille1.fr/. We thank NASA for providing the MODIS and CALIPSO data, which are available at https://ladsweb.modaps.eosdis.nasa.gov/ and https://asdc.larc.nasa.gov/data/CALIPSO/. We thank the CloudSat Data Processing Center for providing the MODIS-AUX data at https://www.cloudsat.cira.colostate.edu/data-products/modis-aux/. We also thank the AERONET project at NASA/GSFC for providing the ground-based aerosol data. The laboratory experiments to retrieve the dust refractive indices in Di Biagio et al. (2017) that feed this work had received funding from the European Union's Horizon 2020 research and innovation program through the EUROCHAMP–2020 Infrastructure Activity under grant agreement no. 730997. They were supported by the French national programme LEFE/INSU (Les Enveloppes Fluides et l'Environnement/Institut National des Sciences de l'Univers) and by the OSU–EFLUVE (Observatoire des Sciences de l'Univers–Enveloppes Fluides de la Ville à l'Exobiologie) through dedicated research funding to the RED-DUST project. The authors acknowledge the CNRS–INSU for supporting the CESAM chamber as national facility as part of the French ACTRIS Research Infrastructure as well as the AERIS datacenter (https://en.aeris-data.fr/) for distributing and curing the data produced by the CESAM chamber through the hosting of the EUROCHAMP datacenter (https://data.eurochamp.org). The views and conclusions contained in this document are those of the authors and should not be interpreted as representing the official policies, either expressed or implied, of the Army Research Laboratory or the US Government.



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
