# Peer review of "Thermal infrared dust optical depth and coarse-mode effective diameter over oceans retrieved from collocated MODIS and CALIOP observations"

_EGUsphere, 2023_

## Referee Comment (RC1)

Review of "***Thermal infrared dust optical depth and coarse-mode effective diameter retrieved from collocated MODIS and CALIOP observations***" by Zheng et al.

General Comments:

This study developed a novel algorithm to retrieve dust optical depth at 10 µm ($DAOD_{10µm}$) and coarse-mode dust effective diameter ($D_{eff}$) from the collocated MODIS thermal infrared (TIR) products and CALIOP dust vertical profiles over the ocean. The $DAOD_{10µm}$ retrievals are validated against $DAOD_{10.6µm}$ from the combined IIR and CALIOP observations and compared with LMD and ULB IASI DAOD products and have shown improved performance than $DAOD_{10.6µm}$ and high correlations with IASI retrievals. The derived $D_{eff}$ is evaluated by comparing with in-situ measurements from AER-D, SAMUM-2 and SALTRACE field campaigns. Finally, the climatological (2013–2017) distribution of $D_{eff}$ is examined for major dust transport pathways over the North Atlantic, Indian Ocean and North Pacific. The paper is quite well written and very comprehensive, with a clear motivation and thorough background review, detailed methodology and solid analysis, along with discussions of uncertainties. The derived MODIS-CALIOP $DAOD_{10µm}$ adds to the existing TIR DAOD products, and the distribution of $D_{eff}$ over the global ocean provides insights into transport patterns of coarse mode dust. I have some minor suggestions for the authors to consider.

Specific Comments:

1. The evaluations of $D_{eff}$ are mainly through case studies during three field campaigns over the Atlantic basin. While the limitation of not including evaluations for the Pacific Ocean is briefly mentioned in Section 6 (line 830), it probably would be more informative to explain why the validation focuses over the North Atlantic earlier in the data and method section or the beginning of section 4.

2. While validating $DAOD_{10µm}$ against a previously well-validated $DAOD_{10.6µm}$ product (Zheng et al. 2022) is probably sufficient, it is not clear why a direct comparison with AERONET station data as by Song et al. (2021) and Zheng et al. (2022) is not performed. Is this due to smaller sample sizes in a shorter time period (5 years)? Would be good to add some explanation.

3. It would be nice to show a spatial distribution of derived $DAOD_{10µm}$ as well, e.g., similar to that of $D_{eff}$ in Fig. 11.

4. Are $DAOD_{10µm}$ and $D_{eff}$ retrieved for both daytime and nighttime overpasses? Are there any noticeable differences in data quality between daytime and nighttime products?

5. Since both polluted dust and dusty marine aerosols from CALIOP are used (Table 1), will this contribute to the uncertainties of $D_{eff}$ estimation?

6. Line 195, the retrieval focuses over 2013–2017, then why are AMSR-E products "(ceased operation in December 2011)" needed?

7. Line 434, "dust plume is concentrated around 3 km to 4 km (see Figure 3a). Therefore, the HYSPLIT back trajectories are initiated at 3 km and 4 km." However, in Fig. 4, back trajectories are initiated around 2.5 ~ 3 km, if I understand correctly.

8. Line 565, "vertical dust distribution observed by MODIS-CALIOP", are any vertical profiles retrieved as well?

9. Line 645 and Fig. 10 caption (line 650), "seasonal mean", of which season? Do you refer to all the seasons of a year?

10. Fig. 10, consider adding information of RMSE.

Technical Corrections:

Line 24-25, PSD definition should be moved to line 24.

Line 403, can you please add a reference for AERONET?

Fig. 7, why is there a clear boundary between data below and above 8 km in CALIOP total attenuated backscatter plots?

Line 662, add "with IASI-ULB" after "(including NP)".

Fig. 13, why do histogram plots show $DAOD_{10\mu m} > 0.1$ while seasonal mean showing $DAOD_{10\mu m} > 0.005$. Would it be better if both figures use the same criterion to display $DAOD_{10\mu m}$ ?

Fig. 14, figure caption indicates "$DAOD_{10\mu m} > 0.1$ " but plot titles show "$DAOD_{10\mu m} > 0.05$" and "$DAOD_{10\mu m} > 0.005$".

---

## Referee Comment (RC2)

The authors developed a good algorithm to retrieve the coarse-mode dust AOD and particle size using thermal infrared measurements from MODIS and vertical distribution from CALIOP. The retrieval results were validated using ground-based and aircraft measurements to evaluate the algorithm reliability, and were applied in the first climatology study about dust particle size global distribution. I think this new product can provide important data source for global model simulations and radiative impact estimations of dust, so it deserves to be published and can receive more attentions from the research community in the future. The manuscript is well organized and presented, and most figures show good quality. I only have minor comments to make the descriptions clearer before it can be published.

1. Line 24-25: The full name of PSD should be put in the place where it shows up in the first time. When PSD was mentioned, I think it's better to mention the distribution function assumption, such as lognormal distribution.

2. Line 28: Suggest to be "the two DAOD retrievals" to avoid misunderstanding.

3. Line 85: This sentence could become shorter to remove repeated information.

4. Line 100-101: "atmospheric window channels that are ..." $--$ > "atmospheric window channels with little gas absorption."

5. Section 3.1.2: 1) Several RI database were mentioned here. I think some missing information about the similar or different wavelength dependence of RI in three thermal infrared bands from these database is important.
   2) When creating the LUT for spheroidal dust using T-matrix method, how to choose the aspect ratio values? Since this parameter is not retrieved, is a fixed value or an assumed distribution used in the radiative transfer simulation?

6. Figure 2: Small comments about the figure quality: 1) For a), the text for small $D_{eff}$ is too dense and cannot be recognized clearly. 2) For b and c, could you change the colors of the lines with different $D_{eff}$? It is a little hard to follow the change of $D_{eff}$ between these lines. Maybe think about using some colormaps. 3) For b and c, why do you use the $11\mu$m as a reference wavelength for beta ratio? It seems in the main text, you mainly discuss the $BTD_{8-12}$ and $BTD_{11-12}$ and it is a little hard to find the beta ratio between 8 and 12 $\mu$m.

7. Since all the case studies and climatology analysis shown in this manuscript are only over ocean, I suggest to emphasize this "ocean" application in the title. As the authors mentioned, this algorithm can be easily used over land by replacing surface data using land surface, but the retrieval accuracy and uncertainties over land may differ from over ocean and need more studies.

8. One limit of this algorithm is that the $D_{eff}$ is assumed the same at different altitudes. Could the particle size information from CALIOP aerosol extinction profiles at two

wavelengths (532 nm and 1064 nm) help provide the vertical distribution of particle size? Is it possible to add this information in the algorithm? Some discussions and potential studies about this can be mentioned in the last section.

---

## Author Comment (AC1)

Responses to reviewers' comments

We appreciate both reviewers for carefully reading our paper and providing insightful and positive feedback, which helped us improve the manuscript. The major changes to the manuscript are listed below.

1. We clarified the data quality in daytime and nighttime at the beginning of Section 3 and added a specific description of the refractive index (RI) wavelength dependence in Section 3.2.

2. We added the explanations of why we validate the $D_{eff}$ retrievals with the three field campaigns over the Atlantic to the beginning of Section 4.

3. We added the analysis of the seasonal global distribution of $DAOD_{10\mu m}$ with the comparison to the two IASI-based $DAOD_{10\mu m}$ to Section 5.1.

4. We addressed the rest of the minor suggestions from the two reviewers accordingly throughout the manuscript.

Please find our point-by-point response to the referee's comments and the corresponding changes we made to the manuscript below (Comments in black and responses in blue). We believe these revisions have adequately addressed the reviewer's comments and welcome any additional suggestions or comments from the referees and editor.

Reply to Reviewer 1:

General Comments: This study developed a novel algorithm to retrieve dust optical depth at 10 µm ($DAOD_{10\mu m}$) and coarse-mode dust effective diameter ($D_{eff}$) from the collocated MODIS thermal infrared (TIR) products and CALIOP dust vertical profiles over the ocean. The $DAOD_{10\mu m}$ retrievals are validated against $DAOD_{10.6\mu m}$ from the combined IIR and CALIOP observations and compared with LMD and ULB IASI DAOD products and have shown improved performance than $DAOD_{10.6\mu m}$ and high correlations with IASI retrievals. The derived $D_{eff}$ is evaluated by comparing with in-situ measurements from AER-D, SAMUM-2 and SALTRACE field campaigns. Finally, the climatological (2013–2017) distribution of $D_{eff}$ is examined for major dust transport pathways over the North Atlantic, Indian Ocean and North Pacific. The paper is quite well written and very comprehensive, with a clear motivation and thorough background review, detailed methodology and solid analysis, along with discussions of uncertainties. The derived MODIS-CALIOP $DAOD_{10\mu m}$ adds to the existing TIR DAOD products, and the distribution of $D_{eff}$ over the global ocean provides insights into transport patterns of coarse mode dust. I have some minor suggestions for the authors to consider.

Specific Comments:

1. The evaluations of $D_{eff}$ are mainly through case studies during three field campaigns over the Atlantic basin. While the limitation of not including evaluations for the Pacific Ocean is briefly mentioned in Section 6 (line 830), it probably would be more informative to explain why the

validation focuses over the North Atlantic earlier in the data and method section or the beginning of section 4.

Reply: Thank you for your suggestion. The reason why we do not include evaluations of $D_{eff}$ for the North Pacific and the Indian Ocean is that most of the dust-aerosol-focused field campaigns took place in North Africa and North Atlantic. At the same time, there are limited in-situ measurements of dust PSD over the Indian Ocean and North Pacific, such as Li et al. (2000) Quinn et al. (2002) and Clarke et al. (2004), which all took place before the launch of CALIOP in June 2006. Additionally, due to the narrow spatial coverage of CALIOP orbit tracks (i.e., 70 m cross-track footprint diameter (Winker et al., 2010)), it is difficult to find cases in our retrievals that can be well-collocated with the North Pacific in-situ measurements in space and time. Therefore, at this stage, we cannot provide validation for $D_{eff}$ retrieval over the Indian Ocean and North Pacific. However, the validation can be implemented if there are more data of in-situ measured dust particle size available in the future.

We added the corresponding explanations to the beginning of Section 4 from Lines 406 to 413.

2. While validating $DAOD_{10\mu m}$ against a previously well-validated $DAOD_{10.6\mu m}$ product (Zheng et al. 2022) is probably sufficient, it is not clear why a direct comparison with AERONET station data as by Song et al. (2021) and Zheng et al. (2022) is not performed. Is this due to smaller sample sizes in a shorter time period (5 years)? Would be good to add some explanation.

Reply: Thank you for your question. Firstly, comparing $DAOD_{10\mu m}$ with AERONET AOD requires the conversion of DAOD from TIR to VIS, which is subject to the extra uncertainties from the pre-assumed TIR-to-VIS DAOD ratios. In addition, implementing a pixel-by-pixel comparison with AERONET is more challenging in our case as CALIOP has too limited spatial coverage to provide enough AERONET-collocated samples. We have added the corresponding explanations at the beginning of Section 5.1 from Lines 602 to 606.

3. It would be nice to show a spatial distribution of derived $DAOD_{10\mu m}$ as well, e.g., similar to that of $D_{eff}$ in Fig. 11.

Reply: Thank you for your suggestion. We added the analysis of the seasonal global distribution of $DAOD_{10\mu m}$ with the comparison to the two IASI-based $DAOD_{10\mu m}$ to Section 5.1 from Lines 669 to 695.

4. Are $DAOD_{10\mu m}$ and $D_{eff}$ retrieved for both daytime and nighttime overpasses? Are there any noticeable differences in data quality between daytime and nighttime products?

Reply: That is a good question. Yes, we retrieve $DAOD_{10\mu m}$ and $D_{eff}$ in both daytime and nighttime. Yet there are no significant day-night differences of $DAOD_{10\mu m}$ and $D_{eff}$ regarding data quality.

First of all, it should be noted that CALIOP has relatively smaller signal-to-noise ratios in the daytime compared with nighttime due to the solar contamination on the Lidar signal. However, as we applied strict criteria for selecting the high-quality cloud-free dust aerosol profiles described in Appendix B, the data quality in both daytime and nighttime is assured to be at the same level.

[Figure]

**Figure R1: The five-year averaged DAOD$_{10\mu m}$ (a, c) and D$_{eff}$ (b, d) in daytime (a, b) and nighttime (c, d) and their corresponding day-night differences (e, f).**

In addition, we also investigate the day-night difference of DAOD$_{10\mu m}$ and D$_{eff}$ in the five-year average, as shown in Figure R1. We found insignificant day-night difference over oceans for both DAOD$_{10\mu m}$ and D$_{eff}$, although there are slightly larger values (<10%) of DAOD$_{10\mu m}$ and D$_{eff}$ in daytime than that in nighttime (see Figure R1e and R1f) near the source regions, which is reasonable as the diurnal variations of dust emission was more significant over source regions (Qin et al 2023). Therefore, we do not show this result as it is insignificant.

Yet we added the clarification that the data quality in daytime and nighttime are at the same level at the beginning of Section 3 from Lines 242 to 246 as
"It should be noted that CALIOP has relatively smaller signal-to-noise ratios during daytime than nighttime, owing to the influence of solar contamination on the Lidar signal (McGill et al., 2007). Nevertheless, by applying identical selection criteria for high-quality cloud-free dust profiles in both daytime and nighttime, we can ensure that the data quality of the selected CALIOP cloud-free dust profiles remains consistent across both periods."

5. Since both polluted dust and dusty marine aerosols from CALIOP are used (Table 1), will this contribute to the uncertainties of D$_{eff}$ estimation?
Reply: Thank you for the good question. As the feature mask of aerosol subtype from CALIOP are only used for excluding suspicious dust detection and are not used in the retrieval (see

descriptions from Line 968 to 973 in Appendix B), it will not contribute to the uncertainties of $D_{eff}$ estimation.

The dust vertical distribution is derived using the fraction of the pre-assumed dust and non-dust depolarization ratio (DPR) to the observed particulate DPR, described in detail in Appendix B. This step is to estimate the dust vertical distribution from those profiles that have both dust and dust mixture (e.g., polluted dust and dusty marine) in a column.

However, the DPR-based method is likely to include the contribution of sea salt over open oceans and generate non-zero $DAOD_{532nm}$ even without dust. The possible reason is that sea salt would have DPR close to dust when its relative humidity is low (e.g., < 50%), especially in daytime (Haarig et al., 2017). Therefore, we further use the CALIOP vertical feature masks to exclude suspicious dust detections when the profile has no dust, polluted dust and dusty marine layers in the column.

6. Line 195, the retrieval focuses over 2013–2017, then why are AMSR-E products "(ceased operation in December 2011)" needed?
Reply: Thank you for your question. The retrieval aims to operate on the collocated MODIS-CALIOP data from 2007 to 2017, with an observational gap from Aug 2011 to June 2012 due to the transition from AMSR-E to AMSR2. For retrievals from Jan 2007 to Aug 2011, we need to use the AMSR-E sea surface temperature (SST) product. In this study, although we choose to perform the climatological analysis on the retrieval from 2013 to 2017, we do provide the retrieval product from 2007 to 2017. In addition, for the dust case in Jan 2008 presented in Section 4.2.1, we performed the retrieval using the AMSR-E SST product.

To clarify, we added the corresponding descriptions in Section 2.2 from Lines 194 to 196 as

"Specifically, the SST products from AMSR-E and AMSR2 are used for retrievals before August 2011 and after June 2012, respectively, while there will be no retrievals during the observational gap between ASMR-E and ASMR2."

7. Line 434, "dust plume is concentrated around 3 km to 4 km (see Figure 3a). Therefore, the HYSPLIT back trajectories are initiated at 3 km and 4 km." However, in Fig. 4, back trajectories are initiated around 2.5 ~ 3 km, if I understand correctly.
Reply: Thank you for pointing it out. We corrected it from Lines 448 to 449 as

"Note that the vertical distribution of the dust plume is concentrated around 2 km to 4 km (see Figure 3a). Therefore, the HYSPLIT back trajectories are initiated at 2.5 km and 3 km."

8. Line 565, "vertical dust distribution observed by MODIS-CALIOP", are any vertical profiles retrieved as well?
Reply: Yes, we retrieved both the column integrated $DAOD_{10um}$ and the vertically resolved extinction coefficients at 10 μm inferred by the CALIOP dust vertical distribution. We added this information at the end of Section 3 from Lines 401 to 403 as

"Both the column integrated $DAOD_{10um}$ and the vertically resolved extinction coefficients at 10 μm inferred by the CALIOP dust vertical distribution are provided in our retrieval".

9. Line 645 and Fig. 10 caption (line 650), "seasonal mean", of which season? Do you refer to all the seasons of a year?
Reply: Sorry for the confusion. Yes, we refer to all four seasons of a year, and we make corresponding revisions in Line 705 and Fig. 10 (now Fig. 11) in the revised manuscript.

10. Fig. 10, consider adding information of RMSE.
Reply: Thank you for your suggestion. We added the RMSE in Fig. 10 (Now is Fig. 11 in the revised manuscript.)

Technical Corrections:

Line 24-25, PSD definition should be moved to line 24.
Corrected.

Line 403, can you please add a reference for AERONET?
References added.

Fig. 7, why is there a clear boundary between data below and above 8 km in CALIOP total attenuated backscatter plots?
Reply: That is due to the different vertical resolutions of CALIOP total attenuated backscatter at 532 nm between above and below 8.2 km. From -0.5 km to 8.2 km, the vertical resolution is 30 m, while the vertical resolution is 60 m from 8.5 km to 20.2 km (Winker et al., 2004).

Line 662, add "with IASI-ULB" after "(including NP)".
Corrected.

Fig. 13, why do histogram plots show $DAOD_{10\mu m}$ >0.1 while seasonal mean showing $DAOD_{10\mu m}$ > 0.005. Would it be better if both figures use the same criterion to display $DAOD_{10\mu m}$?
Reply: Sorry for the confusion. Due to the relatively large retrieval uncertainty for samples with low $DAOD_{10\mu m}$ (e.g., $DAOD_{10\mu m}$ < 0.1), we present the histograms for the selected samples that have retrieved $DAOD_{10\mu m}$ > 0.1. On the other hand, because the seasonal mean $D_{eff}$ is a climatological mean value, we use the corresponding seasonal mean value of $DAOD_{10\mu m}$ for the masking, which is used in Figure 11 (now Figure 12). To avoid confusion, we remove the mask of the seasonal mean $D_{eff}$ for the histogram plots in Figures 13 and 14 (now Figures 14 and 15).

Fig. 14, figure caption indicates "$DAOD_{10\mu m}$ >0.1 " but plot titles show "$DAOD_{10\mu m}$ >0.05" and "$DAOD10\mu m$ >0.005".
Corrected to be "$DAOD_{10\mu m}$ > 0.05".

Reviewer 2

The authors developed a good algorithm to retrieve the coarse-mode dust AOD and particle size using thermal infrared measurements from MODIS and vertical distribution from CALIOP. The retrieval results were validated using ground-based and aircraft measurements to evaluate the algorithm reliability, and were applied in the first climatology study about dust particle size global distribution. I think this new product can provide important data source for global model simulations and radiative impact estimations of dust, so it deserves to be published and can receive more attentions from the research community in the future. The manuscript is well organized and presented, and most figures show good quality. I only have minor comments to make the descriptions clearer before it can be published.

1. Line 24-25: The full name of PSD should be put in the place where it shows up in the first time. When PSD was mentioned, I think it's better to mention the distribution function assumption, such as lognormal distribution.
Reply: Thank you for your suggestions, we corrected it as suggested.

2. Line 28: Suggest to be "the two DAOD retrievals" to avoid misunderstanding.
Reply: Corrected as suggested.

3. Line 85: This sentence could become shorter to remove repeated information.
Reply: Corrected as "Therefore, using TIR observation has an inherent advantage of directly retrieving DAOD without contributions from other aerosols."

4. Line 100-101: "atmospheric window channels that are ..." —— > "atmospheric window channels with little gas absorption."
Reply: Corrected as "…atmospheric window channels most sensitive to dust aerosols with little gas absorption".

5. Section 3.1.2: 1) Several RI database were mentioned here. I think some missing information about the similar or different wavelength dependence of RI in three thermal infrared bands from these database is important.
Reply: Thank you for the suggestion. In this study, we use one RI database from Di Biagio et al (2017) that has nineteen regional RIs in different source regions. The wavelength dependence of RI in the three thermal infrared window channels is a result of different dust compositions in different source regions. Figure R2 (added to be Figure S7 in the supplement) presents the wavelength dependence of RI in terms of $\beta$-ratios. In Figure R2a, the $\beta$-ratio from 12 μm to 11 μm has limited sensitivity to the change of dust RIs, while in Figure R2b, the $\beta$-ratio from 12 μm to 8.5 μm has noticeable variations corresponding to dust RIs. The greater sensitivity of the $\beta$-ratio from 12 μm to 8.5 μm further reshapes the look-up tables of $BTD_{8-12}$ and leads to the results in Figures S8 and S9 in the supplemental materials.

[Figure]

**Figure R2: (a) The β-ratio to 11 μm calculated based on the dust refractive indices from Di Biagio et al. (2017) and D$_{eff}$ = 4.5 μm within the TIR spectrum between 7.5 μm and 13.5 μm. (b) Same as (a) but the β-ratio to 8.5 μm.**

Therefore, we evaluate the uncertainty contributed by different RI assumptions in Section 3.2. The corresponding description of the RI wavelength dependence is added in Section 3.2 from Lines 357 to 359.

2) When creating the LUT for spheroidal dust using T-matrix method, how to choose the aspect ratio values? Since this parameter is not retrieved, is a fixed value or an assumed distribution used in the radiative transfer simulation?
Reply: Good question. The spheroidal aspect ratios follow the fixed size-independent distribution obtained from Dubovik et al. (2006). We first calculate the single particle scattering properties using the T-matrix method. Afterward, the bulk properties are integrated by the single particle properties according to the dust particle size distribution and the fixed aspect ratio distributions. The dust bulk properties finally serve as inputs in the radiative transfer calculation. To clarify that, we revise the descriptions in Section 3.1.2 in Line 309.

6. Figure 2: Small comments about the figure quality: 1) For a), the text for small D$_{eff}$ is too dense and cannot be recognized clearly. 2) For b and c, could you change the colors of the lines with different D$_{eff}$? It is a little hard to follow the change of D$_{eff}$ between these lines. Maybe think about using some colormaps. 3) For b and c, why do you use the 11μm as a reference wavelength for beta ratio? It seems in the main text, you mainly discuss the BTD$_{8-12}$ and BTD$_{11-12}$ and it is a little hard to find the beta ratio between 8 and 12 μm.
Reply: Thank you for the suggestions. We adjusted the font size of the text for D$_{eff}$ in Figure 2a. We changed the colormaps of curves for different D$_{eff}$ in Figures 2b and 2c. In Figures 2b and 2c, the use of the 11 μm beta ratio is to infer the relative changes of the BTD$_{11-12}$. To better inferring BTD$_{8-12}$, we move Figure 2c to be Figure 2b and present the beta ratio to 8.5 μm in Figure 2c. The corresponding changes in text are made from Lines 348 to 355.

7. Since all the case studies and climatology analysis shown in this manuscript are only over ocean, I suggest to emphasize this "ocean" application in the title. As the authors mentioned, this algorithm can be easily used over land by replacing surface data using land surface, but the retrieval accuracy and uncertainties over land may differ from over ocean and need more studies.
Reply: Thank you for your suggestion. We revised the title accordingly.

8. One limit of this algorithm is that the $D_{eff}$ is assumed the same at different altitudes. Could the particle size information from CALIOP aerosol extinction profiles at two wavelengths (532 nm and 1064 nm) help provide the vertical distribution of particle size? Is it possible to add this information in the algorithm? Some discussions and potential studies about this can be mentioned in the last section.
Reply: That is a very good point! Indeed, the CALIOP layer attenuated backscatter total color ratio (i.e., the ratio of the layer total attenuated backscatter at 1064 nm to that at 532 nm) is useful for inferring information about the aerosol particle size. In this study, as the vertical distribution of dust particle size inferred by in-situ measurements has a discernible trend of variation (Ryder et al., 2018), we, therefore, assume the dust particle size distribution to be vertically homogenous.

However, it is a very good suggestion to be considered in future studies. We added the corresponding description in Section 6 from Lines 883 to 886 as

"Thirdly, the vertical distribution of dust PSD in columns is assumed to be homogeneous, which might be improved by inferring the layer attenuated backscatter total color ratio (i.e., the ratio of the layer total attenuated backscatter at 1064 nm to that at 532 nm) observed by spaceborne Lidars."

**Reference:**

Clarke, A. D., Shinozuka, Y., Kapustin, V. N., Howell, S., Huebert, B., Doherty, S., Anderson, T., Covert, D., Anderson, J., Hua, X., Moore II, K. G., McNaughton, C., Carmichael, G., and Weber, R.: Size distributions and mixtures of dust and black carbon aerosol in Asian outflow: Physiochemistry and optical properties, J Geophys Res Atmospheres, 109, https://doi.org/10.1029/2003JD004378, 2004.

Di Biagio, C., Formenti, P., Balkanski, Y., Caponi, L., Cazaunau, M., Pangui, E., Journet, E., Nowak, S., Caquineau, S., Andreae, M. O., Kandler, K., Saeed, T., Piketh, S., Seibert, D., Williams, E., and Doussin, J.-F.: Global scale variability of the mineral dust long-wave refractive index: a new dataset of in situ measurements for climate modeling and remote sensing, Atmos Chem Phys, 17, 1901-1929, 10.5194/acp-17-1901-2017, 2017.

Li, S.-M., Tang, J., Xue, H., and Toom-Sauntry, D.: Size distribution and estimated optical properties of carbonate, water soluble organic carbon, and sulfate in aerosols at a remote high altitude site in western China, Geophys Res Lett, 27, 1107-1110, https://doi.org/10.1029/1999GL010929, 2000.

McGill, M. J., Vaughan, M. A., Trepte, C. R., Hart, W. D., Hlavka, D. L., Winker, D. M., and Kuehn, R.: Airborne validation of spatial properties measured by the CALIPSO lidar, J Geophys Res Atmospheres, 112, https://doi.org/10.1029/2007JD008768, 2007.

Quinn, P. K., Coffman, D. J., Bates, T. S., Miller, T. L., Johnson, J. E., Welton, E. J., Neusüss, C., Miller, M., and Sheridan, P. J.: Aerosol optical properties during INDOEX 1999: Means, variability, and controlling factors, J Geophys Res Atmospheres, 107, INX2 19-11-INX12 19-25, https://doi.org/10.1029/2000JD000037, 2002.

Ryder, C. L., Marenco, F., Brooke, J. K., Estelles, V., Cotton, R., Formenti, P., McQuaid, J. B., Price, H. C., Liu, D., Ausset, P., Rosenberg, P. D., Taylor, J. W., Choularton, T., Bower, K., Coe, H., Gallagher, M., Crosier, J., Lloyd, G., Highwood, E. J., and Murray, B. J.: Coarse-mode mineral dust size distributions, composition and optical properties from AER-D aircraft measurements over the tropical eastern Atlantic, Atmos. Chem. Phys., 18, 17225-17257, 10.5194/acp-18-17225-2018, 2018.

Winker, D., Hunt, W., and Hostetler, C.: Status and performance of the CALIOP lidar, Remote Sens-basel, SPIE2004.

Winker, D. M., Pelon, J., Coakley, J. A., Jr., Ackerman, S. A., Charlson, R. J., Colarco, P. R., Flamant, P., Fu, Q., Hoff, R. M., Kittaka, C., Kubar, T. L., Le Treut, H., Mccormick, M. P., Mégie, G., Poole, L., Powell, K., Trepte, C., Vaughan, M. A., and Wielicki, B. A.: The CALIPSO Mission: A Global 3D View of Aerosols and Clouds, B Am Meteorol Soc, 91, 1211-1230, 10.1175/2010bams3009.1, 2010.